



# Mesoscale nesting interface of the PALM model system 6.0

Eckhard Kadasch[1], Matthias Sühring[2], Tobias Gronemeier[2], and Siegfried Raasch[2]

[1]Deutscher Wetterdienst, Offenbach, Germany
[2]Institute of Meteorology and Climatology, Leibniz University Hannover, Hannover, Germany

**Correspondence:** Eckhard Kadasch (eckhard.kadasch@gmail.com)

**Abstract.** In this paper, we present a newly developed mesoscale nesting interface for the PALM model system 6.0, which enables PALM to simulate the atmospheric boundary layer under spatially heterogeneous and non-stationary synoptic conditions. The implemented nesting interface, which is currently tailored to the mesoscale model COSMO, consists of two major parts: (i) the preprocessor INIFOR, which provides initial and time-dependent boundary conditions from mesoscale model output

and (ii) PALM's internal routines for reading the provided forcing data and superimposing synthetic turbulence to accelerate the transition to a fully developed turbulent atmospheric boundary layer.

We describe in detail the conversion between the sets of prognostic variables, transformations between model coordinate systems, as well as data interpolation onto PALM's grid, which are carried out by INIFOR. Furthermore, we describe PALM's internal usage of the provided forcing data, which besides the temporal interpolation of boundary conditions and removal of

any residual divergence includes the generation of stability-dependent synthetic turbulence at the inflow boundaries in order to accelerate the transition from the turbulence-free mesoscale solution to a resolved turbulent flow. We demonstrate and evaluate the nesting interface by means of a semi-idealized benchmark case. We carried out a large-eddy simulation (LES) of an evolving convective boundary layer on a clear-sky spring day. Besides verifying that changes in the inflow conditions enter into and successively propagate through the PALM domain, we focus our analysis on the effectiveness of the synthetic turbu-

lence generation. By analysing various turbulence statistics, we show that the inflow in the present case is fully adjusted after having propagated for about 1.5 eddy turn-over times downstream, which corresponds well to other state-of-the-art methods for turbulence generation. Furthermore, we observe that numerical artefacts in the form of grid-scale convective structures in the mesoscale model enter the PALM domain, biasing the location of the turbulent up- and downdrafts in the LES.

With these findings presented, we aim to verify the mesoscale nesting approach implemented in PALM, point out specific

shortcomings, and build a baseline for future improvements and developments.

## 1 Introduction

The simulation of urban flows under realistic conditions poses a multiscale problem where evolving synoptic scales interact with building- and street-size scales. While the continuing growth of available computational resources enables large-eddy simulation (LES) to be applied to more and more realistic scenarios at regional scales (Schalkwijk et al., 2015), it still remains

unfeasible to simulate mesoscale processes at resolutions fine enough to represent small-scale turbulence generated by urban structures. To overcome this hurdle and consider synoptically evolving conditions in high-resolution LES models, various





concepts with different degrees of idealization have been developed to couple LES models to larger-scale models. To date, modellers either employ cyclic boundary conditions and add large-scale forcing and nudging terms to the prognostic equations (e.g. Heinze et al., 2017), or they may employ grid-nesting approaches (e.g. Mirocha et al., 2014) with time-dependent in- and outflow boundary conditions.

Both approaches face particular challenges, mainly linked to the representation of the turbulent flow at the domain boundaries, requiring large buffer zones to move boundary effects away from the region of interest. In the first approach, periodic domain boundaries allow unrealistic flow feedbacks due to reentering flow structures caused by complex terrain, urban surfaces, or other surface heterogeneity. Furthermore, feedbacks are not limited to the velocity field. When anthropogenic heat or chemical compounds are emitted, unrealistic thermodynamic conditions or concentrations would re-enter the model domain on

the opposite boundary modifying the upstream conditions for the urban environment, which in turn may bias the distribution of heat and mass concentrations. Here, buffer zones help to move the affected flow region outwards (Letzel et al., 2012; Maronga and Raasch, 2013). Schalkwijk et al. (2015) used a hybrid nesting approach to minimize scalar and mean flow feedbacks from re-entering wakes. They used cyclic boundary conditions in order to retain turbulent fluctuation across the domain but relax horizontal velocities, temperature, and specific humidity towards the parent mesoscale model in a region close to the LES do-

main boundaries. However, since the relaxation only shifts the mean state towards the parent model, turbulent wakes generated by local surface heterogeneities like orography, buildings, etc. may reenter on opposite boundaries nevertheless.

Alternatively, grid-nesting approaches can be employed, which realize a one-way coupling via time-dependent inflow and outflow boundary conditions derived from a larger-scale parent model. In mesoscale models with horizontal grid spacings in the order of $\mathcal{O}(1\,\text{km})$, however, the turbulent exchange of momentum, heat, and water vapor is parametrized so that the their

prognostic fields and derived LES boundary conditions lack turbulent fluctuations. For proper representation of the turbulent flow in the atmospheric boundary layer within the domain of interest, the incoming flow field should be fully spatially developed, i.e. it should not depend on the distance to the inflow boundary layer any more (Lee et al., 2019). This requires buffer zones at the inflow boundaries where turbulence can spatially develop. Mirocha et al. (2014) showed that without adding any perturbations it takes a fetch length of several tens of kilometers to obtain fully spatially developed turbulence, meaning that

significant parts of the computational resources are only spend on the buffer zones.

To reduce the required fetch length, various approaches to generate turbulent inflow conditions exist; for a comprehensive overview about existing methods we refer to Wu (2017). For simulations of atmospheric boundary-layer flows, turbulence recycling approaches are often used (e.g. Park et al., 2015; Munters et al., 2016; Gronemeier et al., 2017). For simulations with one defined inflow boundary, PALM offers a turbulence recycling method according to Kataoka and Mizuno (2002),

where a turbulent signal is read from a defined plane of the model grid and imposed onto the stationary mean profiles at the inflow boundary. To apply this approach, the flow conditions within the recycling plane should be statistically homogeneous, in order to avoid feedbacks between the turbulent signal at the recycling plane and the inflow boundary (Munters et al., 2016). In simulations with realistic land surface distributions, complex terrain or buildings present, however, statistically homogeneous turbulence at the recycling plane cannot be guaranteed without adding large buffer zones. Moreover, due to the evolving

boundary conditions accompanied with changing inflow boundaries, the location of the recycling plane may change and it



is not clear what happens e.g. for diagonal inflows, making the turbulence recycling difficult to apply for evolving inflow conditions.

In contrast to recycling methods, volume-forcing approaches do not necessarily require homogeneous inflow conditions. To accelerate the spatial development of a turbulent flow, Muñoz-Esparza et al. (2014) implemented the so-called cell-perturbation
method into the Weather Research and Forecasting Model (WRF, Skamarock et al., 2008), where box-like perturbations are added onto the potential temperature within a defined region close to the inflow boundary. This was further developed by Muñoz-Esparza et al. (2015) and Muñoz-Esparza and Kosović (2018) by scaling the thermal perturbation amplitude depending on the stability regime. With this approach, Muñoz-Esparza and Kosović (2018) could significantly reduce the required fetch length to obtain fully adjusted turbulence, even under neutral and stable stratifications; though still adjustment fetch lengths up
to about 15 km are required for moderate wind speeds and typical boundary-layer depths. Nevertheless, the cell-perturbation method has shown promising results when applied in nested WRF-LES simulations of a full diurnal cycle for a real-world setup (Muñoz-Esparza et al., 2017), as well as in simulations for ocean-island interactions (Jähn et al., 2016) and coastal sea breeze events (Jiang et al., 2017). Furthermore, prescribing WRF output data as boundary conditions in a PALM simulation, Lee et al. (2019) demonstrated the ability of the cell-perturbation method in a densely built-up urban environment, where
the required buffer zones could be significantly reduced compared to a non-perturbed simulation. However, imposing scaled temperature fluctuations in purely shear-driven or stable boundary layers changes the physics of turbulence generation, with turbulent kinetic energy (TKE) produced by buoyancy rather than shear. Hence, Mazzaro et al. (2019) extended the original cell perturbation method by adding scaled perturbations onto the velocity components at the near inflow region. This approach showed a comparable performance compared to the original version but with a faster spatial development close to the inflow
boundary but a longer adjustment fetch required to achieve an equilibrium state.

An alternative to volume-forcing approaches are so-called filtering approaches, where spatially and temporally correlated perturbations are imposed only onto the velocity components at the lateral boundaries (e.g. Klein et al., 2003; Xie and Castro, 2008). Gronemeier et al. (2015) have originally implemented the synthetic turbulence generation method by Xie and Castro (2008) into PALM and found good agreement of the turbulent flow development in an urban environment compared to using
the turbulence recycling method according to Kataoka and Mizuno (2002). By adding scaled perturbations onto the velocity components the physics of turbulence production are not altered, in contrast to the cell perturbation method, i.e., turbulence can freely develop depending on the mean-gradients of potential temperature and wind speed, making the approach applicable for a wide range of boundary-layer flows as well. The main challenge of this approach, however, is to adequately infer Reynolds-stress components, as well as turbulent length and time scales of the flow to generate appropriate inflow turbulence.
Beside the necessity to add perturbations at the boundaries, modellers should also be aware that numerical artefacts from the mesoscale model may propagate into the LES; e.g. Mazzaro et al. (2017) and Muñoz-Esparza et al. (2017) showed that under-resolved flow structures propagate from a mesoscale WRF simulation into the LES. Honnert et al. (2011) found that in convection-permitting mesoscale simulations, resolved-scale convection on the grid scale can develop when the boundary-layer depth approaches the horizontal grid resolution. When boundary-layer convection is explicitly resolved in mesoscale
models, this is often referred to as the grey zone, or terra incognita (Wyngaard, 2004), where both, mesoscale and LES model





assumptions break, i.e. the grid spacing is already too small compared to the dominant length scales of the flow to justify usage of fully parameterized boundary-layer schemes but still too large to reliably resolve convective structures. Ching et al. (2014) and Zhou et al. (2014) showed that the strength and spatial scale of the resolved-scale convection strongly depends on the horizontal grid resolution, while Shin and Dudhia (2016) also confirmed a dependence on the applied boundary-layer

scheme. With a grid nesting of a turbulence-resolving WRF simulation into a mesoscale WRF simulation, Mazzaro et al. (2017) showed that such under-resolved flow structures propagate into the LES, delaying the spatial development of turbulence. For a strongly convective case, Mazzaro et al. (2017) further showed that first-order statistics in the LES are not significantly affected by imposed under-resolved convection from the parent mesoscale simulation when the flow has been fully adjusted, though variances, turbulent vertical fluxes, and length scales in the LES tend to become larger for stronger under-resolved

mesoscale convection. Further, they showed that the signals of the imposed up- and downdrafts from the mesoscale model vanish after about $40\,\mathrm{km}$ downstream of the inflow boundary, even though they also noted that under less convective conditions the signals may even persist longer. However, this implies that in case of under-resolved convection in the mesoscale model, the turbulent transport in the LES as well as the location where up- and downdrafts occur, depend on the mesoscale model setup, i.e. horizontal resolution, boundary-layer parameterization, etc. In our test scenario we also found under-resolved roll-like

convective structures that propagate into the LES domain.

Another issue that emerges when nesting LES in mesoscale models concerns the representation of the atmospheric boundary layer. Due to different treatment of turbulent transport, i.e. boundary-layer parameterizations in the mesoscale model versus an explicit representation of turbulent eddies in the LES model, the vertical transport of energy, water, and momentum may differ considerably. In situations where this is the case, the mean state of the LES solution, which is generally more credible due

to a wider range of explicitly resolved turbulent scales, will be nudged towards the mesoscale solution including any possible model biases.

In this paper, we present the mesoscale nesting interface of the PALM 6.0 model system. It provides time dependent spatially heterogeneous boundary conditions for PALM obtained from the mesoscale model COSMO (see for instance Baldauf et al., 2011) and includes a stochastic turbulence generation method to accelerate generation of turbulent fluctuations at the

model boundaries. COSMO has been developed by the Consortium for Small-scale Modeling[1] and currently serves as the operational regional weather forecasting model at the German Meteorological Service (Deutscher Wetterdienst, DWD). PALM's mesoscale-nesting interface consists of two major parts: (i) the preprocessor INIFOR, which derives initial and boundary conditions from mesoscale model output, and (ii) PALM's internal routines for reading these forcing data and superimposing synthetic turbulence. In particular, we impose synthetic turbulent structures at the lateral boundaries following Xie and Castro

(2008), while the required turbulence statistics are parameterized based on mesoscale model output. At the moment, INIFOR is tailored towards the COSMO model, but extensions to WRF and ICON (Zängl et al., 2015; Reinert et al., 2020) are planned in the future.

This approach provides a one-way nesting capability of PALM into a mesoscale simulation, where boundary conditions are only set for child model.

---

[1]http://cosmo-model.org





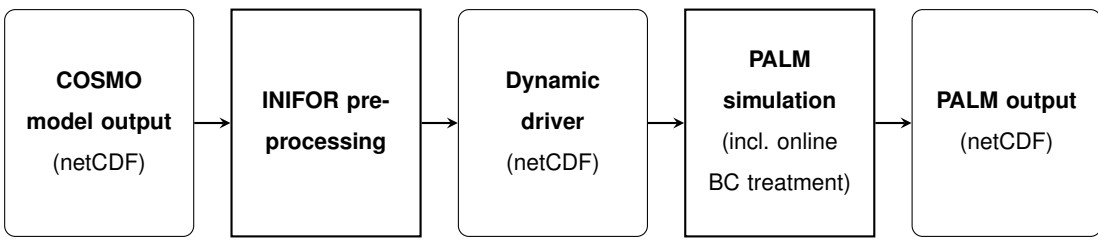

**Figure 1.** Simulation workflow using PALM's mesoscale-nesting interface

At this point, we want to distinguish the mesoscale nesting interface from PALM's self nesting capabilities (Hellsten et al., 2020). While self-nesting allows a two-way coupling of a PALM child domain within a parent PALM domain, the mesoscale nesting interface presented in this paper realizes a one-way or offline nesting of PALM within a mesoscale model. That means, while PALM obtains time-dependent boundary conditions from the mesoscale model, information produced by PALM is not

fed back into the mesoscale model. Both nesting features may, however, may be combined to carry out LES nested in COSMO with one or multiple two-way coupled child nests within PALM.

This paper is structured as follows. We describe the mesoscale nesting approach in Sect. 2, including the relevant mesoscale-microscale model differences, the resulting transformation and interpolation methodology implemented in INIFOR, as well as the synthetic inflow turbulence generation method with its underlying turbulence parametrizations. We demonstrate and verify

our mesoscale nesting approach in a semi-idealized benchmark simulation of a convective boundary layer under evolving synoptic conditions. We describe the setup in Sect. 3 and analyse the case thereafter in Sect. 4. We conclude this paper with a summary of our findings and an outlook to future developments in Sect. 5.

## 2   Mesoscale nesting interface

The PALM model is nested into the mesoscale model by prescribing initial conditions and time-dependent Dirichlet boundary

conditions derived from output of the parent mesoscale model. Boundary conditions for PALM are given for the top and lateral domain boundaries. The boundary conditions at the surface are provided by PALM's urban- and land-surface model, which are initialized from the mesoscale data.

The boundary conditions enter PALM via the mesoscale-nesting interface, which consists of two major components: (i) the pre-processor INIFOR and (ii) PALM's internal boundary condition routines. The workflow of a model run using the

mesoscale-nesting interface is illustrated in Fig. 1. First, the forcing data are interpolated in a pre-processing step using INIFOR and stored in a netCDF driver file. In analogy to the *static* driver (Maronga et al., 2020), which contains all static geospatial information such as topography, building and surface parameters, etc., we refer to this forcing file as the *dynamic driver*. During the simulation, PALM successively reads and processes the dynamic driver data. This includes temporal interpolation of the



**Table 1.** INIFOR's input and output variables. INIFOR supplies initial and boundary conditions for the variables marked with • and initial conditions for variables marked with ○.

| | COSMO database output | | | PALM prognostic variables | | |
|---|---|---|---|---|---|---|
| | Variable | Unit | Symbol | Symbol | Unit | Variable |
| Spherical wind components | $\mathrm{m\,s^{-1}}$ | $U, V, W$ | • $u, v, w$ | $\mathrm{m\,s^{-1}}$ | Cartesian wind components |
| Absolute air temperature | K | $T$ | • $\theta$ | K | Air potential temperature |
| Air specific humidity | $\mathrm{kg\,kg^{-1}}$ | $QV$ | • $q_\mathrm{v}$ | $\mathrm{kg\,kg^{-1}}$ | Air specific humidity |
| Air Pressure | Pa \| hPa | $P \,\vert\, PP$ | | | |
| Soil temperature | K | $TS$ | ○ $t_\mathrm{soil}$ | K | Soil temperature |
| Column-integrated soil moisture | $\mathrm{kg\,m^{-2}}$ | $WS$ | ○ $m_\mathrm{soil}$ | $\mathrm{m^3\,m^{-3}}$ | Volumetric soil moisture |

boundary data, removal of any residual divergence, as well as the superposition of synthetic turbulent fluctuations (see Sect. 2.3).

The required prognostic variables for which the dynamic driver provides initial and boundary conditions are listed in Tab. 1 next to their equivalents in the COSMO model output. Note that we use upper-case letters to denote COSMO's dependent variables and lower-case ones for PALM. In particular, INIFOR provides data for the state of the atmosphere ($u, v, w, \theta$, and $q_\mathrm{v}$) at model start, which can be supplied either as one-dimensional vertical profiles (level-of-detail, LOD = 1) or as three-dimensional fields (LOD = 2). Since the initial atmospheric data is already interpolated onto the PALM's Cartesian grid by INIFOR (see Sect. 2.2), it can be directly copied onto the respective internal PALM arrays after it is read from the dynamic driver. Further, the dynamic driver contains the initial state of soil moisture ($m_\mathrm{soil}$) and temperature ($t_\mathrm{soil}$), again either as one-dimensional profiles (LOD = 1) or as horizontally heterogeneous three-dimensional data (LOD = 2). To allow for different number of soil layers in the PALM domain depending on the local soil type, we decided to linearly interpolate the provided soil data during soil-model initialization rather than in a preprocessing step done by INIFOR as it is done for the initial atmospheric quantities. At this point, we note that the provided initial soil data only contains values aggregated over a mesoscale grid cell, which in reality may feature surfaces with various soil types and different land use across which soil moisture and temperature can vary significantly.

Hence, we recommend to run the soil-model spin-up mechanism as described in Maronga et al. (2020) to obtain individual soil moisture and temperature profiles that are in equilibrium with the local conditions at each model surface. In case of self nesting, where fine-resolution child domains are nested within a coarser-resolution outer parent domain, it is sufficient to provide initial mesoscale model data for the outermost parent domain only, while the respective initial data is propagated to the nested child domains as described in Hellsten et al. (2020). However, the user may also provide a separate dynamic driver for PALM to initialize atmosphere and soil quantities in the respective child domain, which is useful, for instance, if high-resolution initial soil data for a limited area is available.

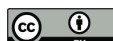



In addition to the initial state, the dynamic driver provides the time-dependent boundary conditions for PALM's atmospheric prognostic variables ($u, v, w, \theta$, and $q_{\mathrm{v}}$) at the top and the four lateral boundaries at certain points in time (hourly data is provided from COSMO output). These time-dependent boundary data are read from the dynamic driver and are linearly interpolated onto the respective model time level, while the data is copied onto the respective model boundaries. In order to

save memory, only the boundary data at LES time levels $t_i$ and $t_{i+1}$ are read, with $t_i \leq t_{\mathrm{s}} < t_{i+1}$, while $t_{\mathrm{s}}$ being the actual simulation time in the model. The boundary data can be provided either as one-dimensional vertical profiles (one value for the top boundary) that are identical at each of the lateral boundaries (LOD = 1), or as individual two-dimensional $x$-$z$ (north and south lateral boundary), $y$-$z$ (east and west boundary) and $x$-$y$ (top boundary) cross-sections.

The velocity boundary conditions and the associated mass-flux fields obtained from a compressible mesoscale model such

as COSMO do generally not satisfy the divergence-free condition of incompressible models such as PALM. To overcome this, we correct the velocity $w_{\mathrm{b}}^{\mathrm{t}}$ at the top boundary similar to the approach described by Hellsten et al. (2020). The correction is calculated from the integrated mass flux through the lateral and top boundaries as

$$w_{\mathrm{c}} = \frac{1}{\rho_0(z_{\mathrm{top}})\,\Omega_{\mathrm{top}}} \int\limits_{\partial\Omega} \rho_0 \boldsymbol{u}_{\mathrm{b}}\boldsymbol{n}\, d\Omega\,, \tag{1}$$

where $\boldsymbol{u}_{\mathrm{b}}$ denotes the velocity vector at the boundary, $\boldsymbol{n}$ the boundary normal vector and $\Omega$ the surface area of the model

boundaries. We obtain divergence-free boundary conditions by using the corrected vertical velocity

$$w_{\mathrm{b}}^{\mathrm{t}}(x,y) = w_{\mathrm{b}}'^{\mathrm{t}}(x,y) + w_{\mathrm{c}}\,, \tag{2}$$

instead of the preliminary boundary condition $w_{\mathrm{b}}'^{\mathrm{t}}(x,y)$ at the top boundary.

## 2.1   Model differences

In the following, we describe the relevant model properties and point out the relevant differences, which yield the conceptual

steps that need to be carried out by INIFOR. Here, we omit in-depth descriptions of both models and refer the reader to additional publications. In particular, more information about the formulation and numerics of COSMO can be found in the model documentation by Doms and Baldauf (2018). Details and verification studies of COSMO-DE – a particular model configuration used at DWD – have been published by Baldauf et al. (2011). For details about the PALM model system, please see the descriptions by Maronga et al. (2015, 2020) and the publications cited therein.

PALM and COSMO differ in a number of ways, between which INIFOR needs to translate in order to derive PALM forcing data. The first difference lies in the physical formulation of the models. COSMO is a non-hydrostatic limited-area atmospheric model. It is based on fully compressible equations, which are formulated in terms of the three spherical wind components, absolute temperature and pressure, density and multiple water phases. PALM, on the other hand, solves incompressible equations for the moist atmosphere, where either the Boussinesq or an anelastic limit of the Navier-Stokes equations may be used. The

model is formulated in terms of the three Cartesian wind components, the potential temperature and the water vapor mixing ratio. The continuity equation in the anelastic and Boussinesq approximations reduces to divergence constraint $\nabla \cdot (\rho \boldsymbol{v}) \equiv 0$. This restriction is not present in COSMO's compressible formulation and this difference is accounted for in PALM's side of



**Table 2.** Model differences between COSMO and PALM

|  | COSMO | PALM |
|---|---|---|
| Formulation | compressible | incompressible (Boussinesq or anelastic) |
| Turbulence | RANS + PBL scheme | LES (energetic part resolved) or RANS |
| Surface representation | spherical | planar |
| Coordinate system | rotated-pole | Cartesian |
| Horizontal grid | structured, equidistant ($^\circ$) | structured, equidistant (m) |
| Vertical grid | fixed, hybrid terrain-following (lower atm.) / horizontally homogeneous (aloft) | fixed, horizontally homogeneous |

the nesting interface by Eqs. (1) and (2). Furthermore, turbulence is fully parameterized in COSMO such that its flow fields are essentially free of turbulent fluctuations. PALM, on the other hand, explicitly resolves the energetic part of the turbulent spectrum.

Secondly, owed to their different domain extents, both models use different approximations of Earth's surface and, as a

result, use different coordinate systems. COSMO represents the planet as a perfect sphere with radius $R = 6371.229$ km and terrain layered on top of it. Consequently, it uses a spherical coordinate system, in particular a *rotated-pole system* as depicted in Fig. 2a. The origin of COSMO's coordinate system is rotated to the region of interest in order to minimize grid heterogeneity. The rotation is defined in terms of the location of the rotated North Pole with the restriction that the prime meridian continues to intersect with Earth's axis of rotation and, thus, with the geographical North and South Pole. In contrast, PALM is designed

as a tool for simulating the atmospheric boundary layer, which implies domain sizes that are small compared to Earth's radius. Hence, Earth's surface is represented as a tangential plane and the governing equations are formulated in a Cartesian frame of reference with the $z$ coordinate aligned with the uniform gravitation vector field and the $y$ coordinate facing north.

Lastly, COSMO and PALM use different numerical grids, which requires interpolation. Both models discretize their respective governing equations on structured grids aligned with their respective coordinate axes and equidistant horizontal spacings

– in case of COSMO equidistant in rotated latitudes and longitudes, and in the case of PALM equidistant in Euclidean length. Both are based on the Arakawa-C-type grid structure, where scalars are defined at the *mass points* at the cell centre and velocity components are staggered one half grid cell. In the vertical, both models allow for grid stretching. COSMO uses a hybrid $z$-coordinates, with levels in the lower region following the terrain and gradually approaching an upper region with horizontally homogeneous spacings. The grid is constructed starting with the staggered velocity points, the so-called *half layers*. The *full*

*layers*, where mass points are located, are defined as the arithmetic mean of two neighbouring half layers. PALM, on the other hand, uses a horizontally uniform grid that may contain both, parts with vertically stretched as well as equidistant grid spacings. With PALM, typical grid spacings are on the order of 100 m to 1 m, while COSMO is designed for horizontal grid spacings on the order of 10 to 1 km.

Currently, INIFOR is designed to process model output of DWD's current operational configuration COSMO-D2 (Baldauf

et al., 2018) and its predecessor COSMO-DE (Baldauf et al., 2014). Both configurations operate on rotated-pole grids with





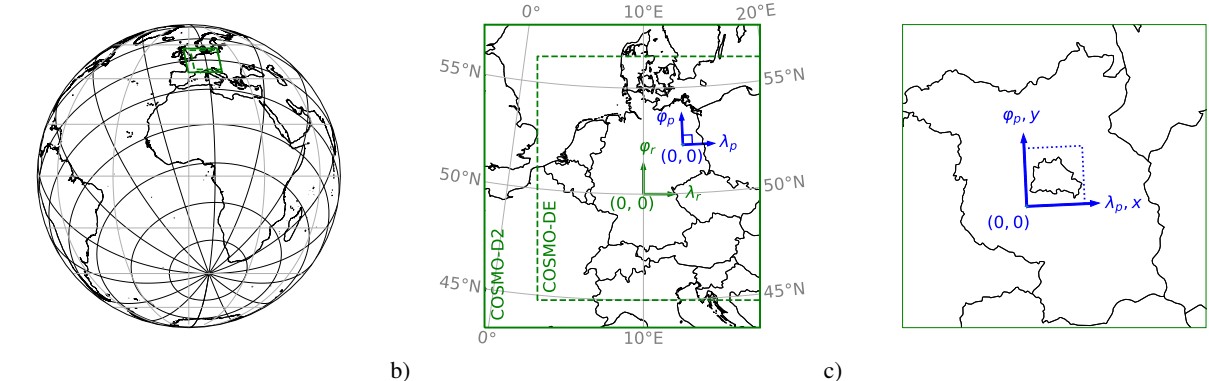

a)                                        b)                                        c)

**Figure 2.** Example PALM domain (blue) for Berlin nested within the DWD COSMO configurations (green). Panel **(a)** Shows the rotated-pole system of COSMO-DE and -D2, the rotated North Poles of which are both located at $(\lambda^N, \varphi^N) = (170°\text{W}, 40°\text{N})$, placing their origin at $(\lambda, \varphi) = (10°\text{E}, 50°\text{N})$ (see panel **(b)**). Panel **(b)** shows the horizontal domain extents of both COSMO configurations. COSMO-D2 extends over $\lambda_r \in [7.5°\text{W}, 5.5°\text{E}], \varphi_r \in [6.3°\text{S}, 8.0°\text{N}]$ (solid green), which is slightly increased compared to the COSMO-DE domain with $\lambda_r \in [5.0°\text{W}, 5.5°\text{E}], \varphi_r \in [5.0°\text{S}, 6.5°\text{N}]$ (dashed green). Panels **(b)** and **(c)** show an example configuration with a PALM domain of $50 \text{ km} \times 50 \text{ km}$ (dashed blue).

the rotated North Pole located at $(\lambda^N, \varphi^N) = (170°\text{W}, 40°\text{N})$, placing the origin at $(\lambda^O, \varphi^O) = (10°\text{E}, 50°\text{N})$, close to the centre of Germany (see Fig. 2b). COSMO-D2 extends the prior COSMO-DE domain slightly towards the north, west and south. With horizontal grid spacings of 2.2 km and 2.8 km, respectively, both configurations run at convection-permitting resolutions (cf. Baldauf et al., 2011). COSMO-D2 uses 65 vertical levels, which is up from 50 levels in COSMO-DE. The

vertical grid spacing of the lowest cell at sea level is 20 m for both configurations, which gets further compressed over elevated terrain. The particular rules governing the vertical grid generation can be found in DWD's database manuals (Baldauf et al., 2011; Baldauf et al., 2018) and in the COSMO model documentation (Doms and Baldauf, 2018), but in the context of data interpolation on that grid, knowledge of the underlying rules is not necessary. Rather, the vertical grid is completely defined by the three-dimensional field of the half-layer heights, which is static in time and available in the model output.

## 2.2 INIFOR preprocessing

PALM and COSMO differ in their physical formulation, i.e. their prognostic and available output variables, the representation of Earth's surface, the coordinate systems, and the structure and resolution of the numerical grids used. To translate those differences, INIFOR needs to carry out the following conceptual steps:

1. convert between the sets of prognostic variables (see Tab. 1),

2. project PALM's Cartesian domain onto COSMO's spherical Earth,

3. transform PALM's projected Cartesian coordinates to COSMO's rotated-pole system,





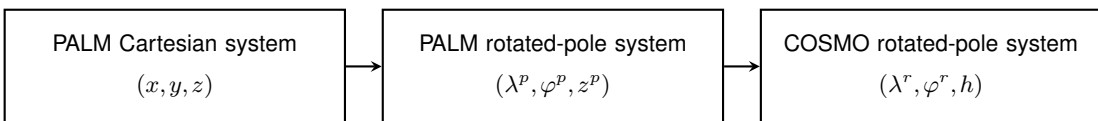

**Figure 3.** Coordinate systems used in INIFOR. The PALM grid coordinates are first projected onto the PALM rotated-pole system (see Eq. (5)), which are then transformed to the COSMO rotated-pole system.

**Table 3.** Coordinate systems and grid indices

| Symbols | Description |
| --- | --- |
| $\lambda, \varphi$ | Geographical longitude and latitude |
| $\lambda^r, \varphi^r, h$ | COSMO rotated longitude, latitude and height above sea level |
| $\lambda^p, \varphi^p, z^p$ | PALM rotated longitude, latitude and height above sea level |
| $x, y, z$ | PALM Cartesian coordinates |
| | |
| $\hat{i}, \hat{j}, \hat{k}$ | COSMO grid point indices |
| $i, j, k$ | PALM grid point indices |

4. interpolate COSMO data onto PALM's grid in COSMO rotated-pole system.

In the following sections, we describe how INIFOR addresses each of these steps in detail.

Note that the data interpolation could be carried out in different coordinate systems. With INIFOR, we decided to interpolate in COSMO's rotated-pole system where the required interpolation neighbours are located on a rectangular grid leading to
simple and efficient interpolation rules. We obtain the COSMO coordinates for the PALM grid points using a two-step transformation as shown in Fig. 3. First, we project the PALM grid onto COSMO's geoid (see Sect. 2.2.2), resulting in a rotated-pole system similar to COSMO's but with a different rotated North Pole. Then, we transform from the rotated PALM system to the rotated COSMO system.

### 2.2.1 Conversion of prognostic variables

Differences in the model formulations require conversions between the sets of prognostic equations. In our case, this includes the computation of the potential temperature and the volumetric soil moisture. INIFOR converts both quantities before interpolating them onto the PALM grid.

As for the air temperature preprocessing, INIFOR replaces the absolute temperature $T$ provided in the COSMO output by the potential temperature given by

$$\theta = T \left( \frac{P}{p_{\mathrm{ref}}} \right)^{R_d/c_p} .$$
(3)





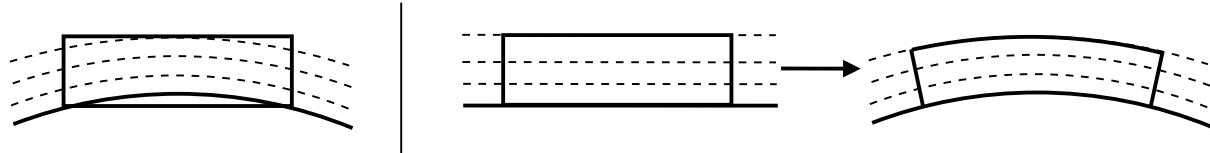

**Figure 4.** Schematic comparison of direct spherical-to-Cartesian transformation (left) and the inverse plate carrée projection (right). The schematic shows a vertical cut through the atmosphere and Earth's surface (solid arc). The solid boxes represent the PALM domain and dashed arcs indicate isosurfaces of Earth's gravitational potential.

Here, $P$ is the corresponding mesoscale pressure, $p_{\mathrm{ref}} = 10^5$ Pa is the reference pressure, and $R_d = 287$ J kg$^{-1}$K$^{-1}$ and $c_p = 1005$ J kg$^{-1}$K$^{-1}$ are the ideal gas constant and specific heat at constant pressure of dry air, respectively.

For soil data, preprocessing is slightly more involved because on sea or inland water cells, COSMO's soil data are not defined. Due to the coarser grid resolution shorelines or inland lakes do not necessarily correspond to the high-resolution

surface input required by PALM. In order to provide soil data at each PALM grid point, the missing information is iteratively generated by horizontal averaging of soil data from neighboring land cells. Every iteration of this procedure generates new virtual land cells. By repeating this procedure using both the original and newly generated virtual cells, virtual shoreline moves one COSMO cell per iteration. This procedure is currently repeated five times, before the field is used for interpolation. After the data extrapolation on the COSMO grid, the units of COSMO's soil moisture are converted to PALM's formulation. COSMO

provides soil moisture as vertically integrated water density while PALM requires the volumetric water content. The conversion is given by

$$m_{\mathrm{soil},k} = \frac{WS_k}{\rho_w \, \Delta d_k}, \tag{4}$$

where $WS_k$ and $\Delta d_k$ are the column-integrated soil moisture and thickness of the $k$-th soil layer in COSMO, respectively, and $\rho_w = 1000$ kg m$^{-3}$ is the approximate density of water.

**2.2.2   Inverse map projection**

There are multiple ways as to how the differences in the representation of Earth's surface can be resolved. Two options are illustrated in Fig. 4: (i) by a direct spatial transformation between coordinates of the rotated-pole coordinates and the tangential Cartesian system, or (ii) by projecting the Cartesian system onto COSMO's geoid surface. The first approach, however, implies a change in the gravitational field: While the isosurfaces of the gravitational potential are concentric spheres (gravitation

vectors point towards the geoid center), they are parallel planes in PALM's Cartesian system. As a result, a balanced stratified atmosphere in COSMO would produce baroclinic instabilities if it was directly transformed into PALM's Cartesian system. With INIFOR, we avoid this effect by choosing the second approach and project PALM's system onto COSMO's geoid. This corresponds to the inverse of a map projection.





In particular, we use an inverse plate carrée projection which linearly maps between the Cartesian coordinates $(x, y, z)$ and the spherical coordinates $(\lambda^p, \varphi^p, z^p)$ on a sphere of Radius $R$ according to

$$\lambda^p(x) = \frac{x}{R}, \quad \varphi^p(y) = \frac{y}{R}, \quad z^p(z) = z, \tag{5}$$

where the superscript $p$ refers to PALM. This projection defines a rotated-pole system, the equator and prime meridian of which pass through PALM's Cartesian origin (see Fig. 2c). By requiring the $y$-axis to point towards geographical North, we obtain a rotated-pole system of the same kind as COSMO's rotated-pole system where the prime meridian also intersects with the Earth's North Pole.

### 2.2.3 Coordinate transformation

When transforming the PALM to the COSMO rotated-pole coordinates, we consider the PALM system a rotated-pole system relative to the COSMO rotated-pole system, the same way the COSMO system is a rotated-pole system relative to the geographical system. Because, as we discuss below, the definition and evaluation of the transformation from PALM's to COSMO's coordinates involves forward and backward transformations between rotated systems, we begin with the general forward and backward transformations. The forward transformation from a geographical $(\lambda, \varphi)$ to a rotated-pole system $(\lambda^r, \varphi^r)$ is obtained from spherical geometry as (Baldauf et al., 2014)

$$\lambda^r(\lambda, \varphi) = \arctan\left( \frac{-\cos\varphi \sin(\lambda - \lambda_N)}{-\cos\varphi \sin\varphi_N \cos(\lambda - \lambda_N) + \sin\varphi \cos\varphi_N} \right), \tag{6}$$

$$\varphi^r(\lambda, \varphi) = \arcsin\left( \sin\varphi \sin\varphi_N + \cos\varphi \cos\varphi_N \cos(\lambda - \lambda_N) \right), \tag{7}$$

where $(\lambda_N, \varphi_N)$ are the geographical coordinates of the rotated pole. The inverse transformation is given by

$$\lambda(\lambda^r, \varphi^r) = \lambda_N - \arctan\left( \frac{\cos\varphi^r \sin\lambda^r}{\sin\varphi^r \cos\varphi_N - \sin\varphi_N \cos\varphi^r \cos\lambda^r} \right), \tag{8}$$

$$\varphi(\lambda^r, \varphi^r) = \arcsin\left( \sin\varphi^r \sin\varphi_N + \cos\varphi^r \cos\lambda^r \cos\varphi_N \right). \tag{9}$$

The definition of the PALM rotated-pole system starts with the specification of its origin in terms of its geographical coordinates $(\lambda_O, \varphi_O)$. In order to define the transformation to the rotated COSMO system, we need to translate the PALM origin into the corresponding rotated North Pole in the COSMO system. We do this by first computing the location of the rotated North Pole $(\lambda_N, \varphi_N)$ in the geographical system as

$$
\left.
\begin{aligned}
\lambda_N &= \begin{cases} \lambda_O - \pi \operatorname{sign}(\lambda_O) & \text{if } \varphi^N > 0 \\ \lambda_O & \text{else} \end{cases} \\
\varphi_N &= \frac{\pi}{2} - |\varphi_O|
\end{aligned}
\right\}
\quad \text{for } -\pi \leq \lambda_O \leq \pi \text{ and } -\frac{\pi}{2} \leq \varphi_O \leq \frac{\pi}{2},
\tag{10}
$$

and then, using the forward transformation in Eqs. (6) and (7), we obtain the rotated North Pole coordinates $\lambda_N^r = \lambda^r(\lambda_N, \varphi_N)$ and $\varphi_N^r = \varphi^r(\lambda_N, \varphi_N)$ in COSMO's frame of reference. Now the horizontal transformation between the PALM and COSMO





is fully defined and we can transform PALM rotated-pole coordinates to COSMO's rotated-pole system using the backward transformation in Eqs. (8) and (9) using the PALM coordinates $(\lambda^p, \varphi^p)$ as the rotated-pole coordinates $(\lambda^r, \varphi^r)$ and COSMO's rotated-pole coordinates $(\lambda^r, \varphi^r)$ as the geographical ones $(\lambda, \varphi)$.

Finally, the PALM domain may generally be shifted above sea level by specifying a non-zero domain base $z_0$ in order to
vertically align COSMO and PALM orography. The COSMO heights (above sea level) of the vertical PALM levels at $z^p$ are then given by

$$h(z^p) = z^p + z_0 \,. \tag{11}$$

### 2.2.4 Spatial interpolation

Having the COSMO rotated-pole coordinates for each PALM grid point available, we can interpolate COSMO fields by locating
the appropriate interpolation neighbours and by computing the corresponding interpolation weights. We use the coordinate symbols laid out in Tab. 3 to describe the interpolation methodology. In particular, the COSMO rotated-pole coordinates are denoted by $(\lambda^r, \varphi^r, h)$ while the Cartesian PALM coordinates are $\boldsymbol{x} = (x, y, z)$. Grid points are referenced with the indices $i, j, k$ for the PALM grid while points on COSMO's grid are denoted by an additional hat.

Using this convention, a general interpolation scheme for an arbitrary scalar $s$ on the COSMO grid can be formulated in
terms of the weighted sum of $N_l$ neighbouring values $S$

$$\hat{s}(\boldsymbol{x}_{ijk}) = \hat{s}_{ijk} = \sum_{l=1}^{N_l} \mathcal{W}_{ijk}^l S_{\hat{i}_{ijk}^l, \hat{j}_{ijk}^l, \hat{k}_{ijk}^l} \quad \text{for} \quad l \in \{1, 2, ..., N_l\} \,. \tag{12}$$

Here, the indices $\hat{i}_{ijk}^l, \hat{j}_{ijk}^l$, and $\hat{k}_{ijk}^l$ identify the $l$-th neighbour on the COSMO grid for the PALM grid point at $\boldsymbol{x}_{ijk}$ and $\mathcal{W}_{ijk}^l$ are the corresponding interpolation weights, which satisfy $\sum_{l=1}^{N_l} \mathcal{W}_{ijk}^l = 1$. Since the scalar's values on the mesoscale grid are known, the remaining task is to compute the values of those four fields. In INIFOR, we use bilinear and trilinear interpolation
requiring only the four or eight closest neighbours, respectively, but the approach may be extended to higher-order schemes by including more points. INIFOR separates horizontal and vertical interpolation, which (i) simplifies the treatment of COSMO's terrain-following vertical grid and (ii) enables us to adapt the horizontal scheme to other grid structures in the future, such as the triangular horizontal grid of ICON, the Icosahedral Nonhydrostatic Model. (As of writing this paper, ICON is being used as the operational global weather prediction model at DWD and ICON-LAM – its limited-area model variant – is designated
to supersede COSMO as the regional model. For a description of ICON's grid see for instance the paper by Wan et al. (2013).)

**Two-dimensional horizontal interpolation** In the case of bilinear interpolation, Eq. (12) reduces to two dimensions and we can drop the vertical indices $\hat{k}$ and $k$ and $N_l = 4$. The indices $\hat{i}_{ij}^l, \hat{j}_{ij}^l$ of the four neighbours $l \in \{1, 2, 3, 4\}$ are

$$\hat{i}_{ij}^l = \begin{cases} \hat{I}_{ij} & \text{for} \quad l \in \{1, 2\} \\ \hat{I}_{ij} + 1 & \text{for} \quad l \in \{3, 4\} \end{cases}$$

$$\hat{j}_{ij}^l = \begin{cases} \hat{J}_{ij} & \text{for} \quad l \in \{1, 4\} \\ \hat{J}_{ij} + 1 & \text{for} \quad l \in \{2, 3\} \,. \end{cases} \tag{13}$$



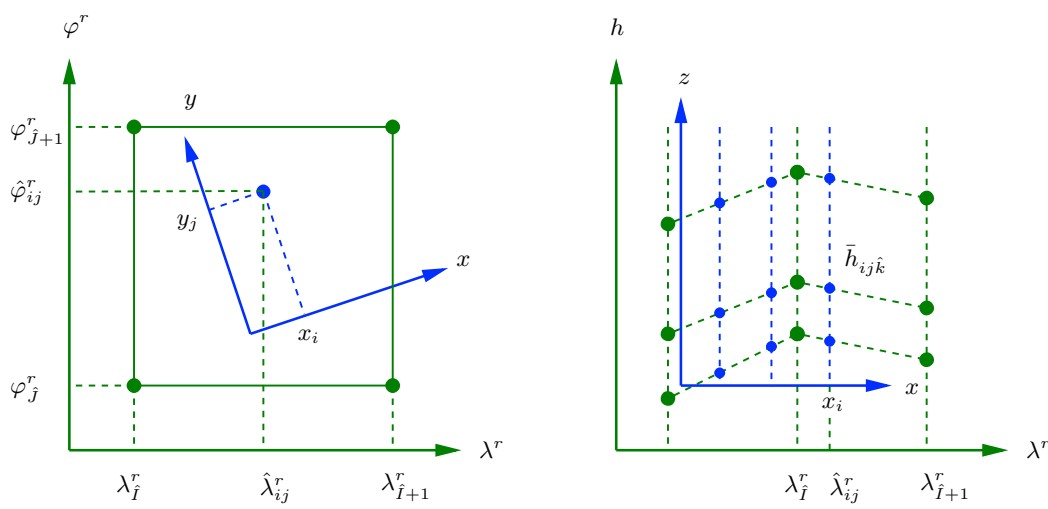

**Figure 5.** Horizontal (left) and vertical cut (right) through an Intermediate PALM grid (blue) within a COSMO rotated-pole grid (green).

The reference COSMO indices $\hat{I}_{ij}$ and $\hat{J}_{ij}$ mark the bottom left neighbour point (see Fig. 5) and are obtained from the rotated-pole coordinates of the PALM grid point according to

$$\hat{I}_{ij} = \text{floor}\left(\frac{\lambda_{i\hat{j}}^r - \lambda_0^r}{\Delta\lambda^r}\right)$$

$$\hat{J}_{ij} = \text{floor}\left(\frac{\varphi_{i\hat{j}}^r - \varphi_0^r}{\Delta\varphi^r}\right),$$

5  where $\lambda_0^r$ and $\varphi_0^r$ mark the lowest longitude and latitude of the COSMO grid and $\Delta\lambda^r$ and $\Delta\varphi^r$ are the equidistant grid spacings in the respective directions.

Using the location of the neighbour grid points, we can compute the corresponding bilinear interpolation weights based on the nondimensional coordinates

$$\zeta_{ij} = \frac{\hat{\lambda}_{ij}^r - \lambda_{\hat{I}}^r}{\lambda_{\hat{I}+1}^r - \lambda_{\hat{I}}^r}, \quad \eta_{ij} = \frac{\varphi_{ij}^r - \varphi_{\hat{J}}^r}{\varphi_{\hat{J}+1}^r - \varphi_{\hat{J}}^r}, \quad \text{with} \quad \zeta_{ij}, \eta_{ij} \in [0,1] \tag{14}$$

10  along the COSMO cells faces. The bilinear interpolation weights at the four neighbour points are given by

$$\mathcal{W}_{ij}^1 = (1 - \zeta_{ij})(1 - \eta_{ij})$$

$$\mathcal{W}_{ij}^2 = (1 - \zeta_{ij})\eta_{ij}$$

$$\mathcal{W}_{ij}^3 = \zeta_{ij}\eta_{ij}$$

$$\mathcal{W}_{ij}^4 = \zeta_{ij}(1 - \eta_{ij}) = 1 - \sum_{l=1}^{3} \mathcal{W}_{ij}^l, \tag{15}$$





which lets us interpolate scalars using to Eq. (12).

**Three-dimensional interpolation**    The interpolation in three dimensions is split in two steps in INIFOR: (i) a bi-linear horizontal interpolation onto an intermediate grid and (ii) a linear vertical interpolation in each of its columns. The intermediate grid, hereafter indicated by an overbar, shares PALM's fine horizontal grid but features COSMO's coarser vertical levels (see Fig. 5). Concretely, the vertical levels $\overline{h}_{i\hat{j}k}$ of the intermediate grid – as well as values of the corresponding interpolation quantity $\overline{s}_{i\hat{j}k}$ – are computed using the bilinear scheme above, i.e.

$$\overline{h}_{ij\hat{k}} = \sum_{l=1}^{4} \mathcal{W}_{ij}^{l} h_{\hat{i}_{ij}^{l}, \hat{j}_{ij}^{l}, \hat{k}} \tag{16}$$

$$\overline{s}_{ij\hat{k}} = \sum_{l=1}^{4} \mathcal{W}_{ij}^{l} S_{\hat{i}_{ij}^{l}, \hat{j}_{ij}^{l}, \hat{k}}, \tag{17}$$

where $h_{ijk}$ are the COSMO grid levels and $l \in [1,4]$ iterates over the four neighbouring COSMO columns. In the second step, the interpolated values $\hat{s}$ are interpolated vertically from the intermediate to the PALM target grid

$$\hat{s}_{ijk} = \sum_{l=1}^{2} \overline{\mathcal{W}}_{ijk}^{l} \overline{s}_{ij\hat{k}_{ijk}^{l}}. \tag{18}$$

**Interpolation of velocities**    The transformation in Eqs. (8) and (9) between the two rotated-pole systems (see Fig. 2b) involves a rotation as the meridians of the original and rotated system are generally not parallel. This deviation angle, the so called *meridian convergence*, increases as we move away from the reference meridian and its distribution is given by Baldauf et al. (2014) as

$$\delta(\lambda_r, \varphi_r) = \arctan\left( \frac{\cos\varphi_r^N \sin(\lambda_r^N - \lambda_r)}{\cos\varphi_r \sin\varphi_r^N - \sin\varphi_r \cos\varphi_r^N \cos(\lambda_r^N - \lambda_r)} \right). \tag{19}$$

We obtain the Cartesian velocity components in the PALM system by rotating COSMO's spherical velocity components by the local meridian convergence according to

$$u = U\cos\delta - V\sin\delta$$
$$v = U\sin\delta + V\cos\delta. \tag{20}$$

Since on the staggered Arakawa-C grid $U$ and $V$ are not defined at the same location, INIFOR first interpolates horizontal velocities onto COSMO's mass points and then rotates the interpolated velocity vectors using Eq.(20). Apart from this preprocessing, velocities are interpolated the same way scalars are. The resulting interpolation neighbours and weights for velocities however do differ from those of scalars because $u$ and $v$ on PALM's staggered grid are in turn defined half a grid cell away from PALM's mass points.

**Averaging of profile data**    INIFOR provides the option to initialize and force PALM with three-dimensional atmospheric data (LOD = 2) or with averaged profiles (LOD = 1). The latter has the advantage that, for large-setups, INIFOR preprocessing is easier to handle in practice because less memory is required on the preprocessing machine and the resulting dynamic driver





is greatly reduced in size because three-dimensional arrays are omitted. INIFOR produces profile data by first averaging along COSMO levels and then interpolating in the vertical direction.

Concretely, this is done carrying out the following steps. We first define the averaging region as a horizontal box bounded by the minimum and maximum rotated longitude and latitude of the PALM domain. Once all COSMO columns in the region

are identified, we compute the average vertical grid levels of the terrain-following COSMO grid and then compute the vertical neighbours and weights for every PALM level relative to the averaged COSMO levels. The average profile (denoted in the following by the double bar) is then formed by scanning through the $N_c$ columns of the averaging region and adding the vertically interpolated values on every PALM level according to

$$\bar{\bar{s}}_{ijk} = \frac{1}{N_c} \sum_{c=1}^{N_c} s_{i_c j_c k} = \frac{1}{N_c} \sum_{c=1}^{N_c} \sum_{l=1}^{2} \bar{\bar{\mathcal{W}}}_k^l S_{\hat{i}_c \hat{j}_c \hat{k}_k^l}, \tag{21}$$

with $(\hat{i}_c, \hat{j}_c)$ being the indices of the $N_c$ COSMO columns in the averaging region.

**A note on large-scale pressure forcing**   As opposed to our comment in the PALM 6.0 Overview paper (Maronga et al., 2020), the geostrophic wind forcing is not required in the present mesoscale nesting interface, which we think, warrants further explanation. In idealized model setups, the assumption of periodicity is often used. However, using periodic boundary conditions prevents the model from developing any mean horizontal pressure gradient. Thus, if large-scale pressure gradients

are important to a given problem, they need to be externally prescribed. This is often done by using an equivalent geostrophic wind profile that is obtained from the mesoscale pressure and density fields $P$ and $\rho$, respectively,

$$u_{1,g} = -\frac{1}{f\rho}\frac{\partial P}{\partial y}, \quad u_{2,g} = \frac{1}{f\rho}\frac{\partial P}{\partial x}, \tag{22}$$

which enters the model in the form of the additional forcing tendency

$$\left[\frac{\partial u_i}{\partial t}\right]_g = \epsilon_{i3j} f u_{g,j}, \quad \text{for } i \in \{1,2\} \tag{23}$$

in the horizontal momentum equations. With this external mesoscale forcing, even an atmosphere initially at rest, will eventually develop a mean horizontal flow representative of the real conditions. In the case of inflow Dirichlet boundary conditions, the situation is reversed: the dynamic pressure develops a mean horizontal gradient as a result of internal forces in order to maintain continuity under the prescribed inflow boundary conditions.

In incompressible formulations, the pressure solution is obtained by constructing and solving a Poisson-type equation which

can be obtained by applying the divergence operator to the momentum equation. The equation is simplified by exploiting the incompressible continuity equation which represents a divergence constraint on the mass flux $\rho v$. As a result, the pressure solution of the Poisson equation acts as to enforce this divergence constraint onto the flow field. In the case when Dirichlet boundary conditions are used for the velocity, any divergence resulting from the tendencies in the momentum equation, will be compensated by a corresponding gradient in the dynamic pressure solution. For instance, mean friction and mean Coriolis

forces will result in pressure gradients opposing those effects.





### 2.2.5 Program structure

INIFOR's program structure is organized around the set of netCDF variables that are to be computed for the dynamic driver. The dynamic driver contains individual netCDF variables for each combination of prognostic variable and model boundary, e.g. netCDF variables for the $u$ velocity component at the east, the south, the west boundary and so forth. Internally, INIFOR

maintains a list with representations of each of those netCDF variables. Each is associated with the netCDF metadata required to handle data input and output, the computational task – averaging profiles or interpolating fields in 2D or 3D – as well as with the appropriate interpolation grids. The latter contain both grid point coordinates and interpolation neighbors and weights. Generally, different variables that are defined at the same grid point type also share the interpolation parameters. For instance, the horizontal (intermediate) interpolation grid for scalars is shared among netCDF variables for the initial soil

moisture and temperature fields as well as the top boundary conditions for $w$, $\theta$ and $q_v$. Consequently, interpolation grids with their corresponding interpolation parameters are computed once and reused each time step and shared among multiple output variables.

This is reflected in INIFOR's program flow, which is depicted in Fig. 6. It is divided into two main sections: a setup phase and the main loop. During the setup phase, INIFOR constructs the required model and interpolation grids. This involves defin-

ing and transforming the coordinates of the PALM interpolation grids as well as precomputing interpolation neighbours and weights for every grid point. During the main loop, INIFOR then iterates through the output netCDF variables and time steps, reusing precomputed interpolation parameters that are associated with each variable. Each main loop iteration is structured into reading input data, preprocessing input data, interpolation, and data output. The preprocessing step is dependent on the kind of input and includes the extrapolation of soil data into water cells, conversions between model formulations (such as the

computation of the potential temperature or the computation of volumetric soil moisture, see Sect. 2.2.1), and velocity vector rotation (see Sect. 2.2.4).

As input data, INIFOR reads hourly netCDF files containing COSMO analyses. Each hourly input is processed separately and translated into one instantaneous boundary condition in the dynamic driver. Input data is not interpolated temporally in INIFOR but rather in PALM during the simulation as described in Sect. 2.

### 25   2.3   Superposition of boundary conditions with synthetic turbulence

With the generation of time-dependent boundary conditions from mesoscale model output in a preprocessing step and online processing of the boundary data, PALM is enabled to simulate more realistic scenarios considering time-evolving synoptic conditions. However, due to the nature of RANS models, turbulence is parametrized and thus the boundary values are free of any turbulent fluctuations. Mirocha et al. (2014) showed that without adding perturbations the turbulent flow needs several tens

of kilometers to sufficiently develop. In order to accelerate the spatial development of turbulence in PALM in our mesoscale nesting approach, we employed the synthetic turbulence generator by Xie and Castro (2008) where perturbations are added onto the $u, v, w$ - components imposed at the lateral boundaries. In the following, the preliminary boundary values without any perturbations added are indicated by an overbar.





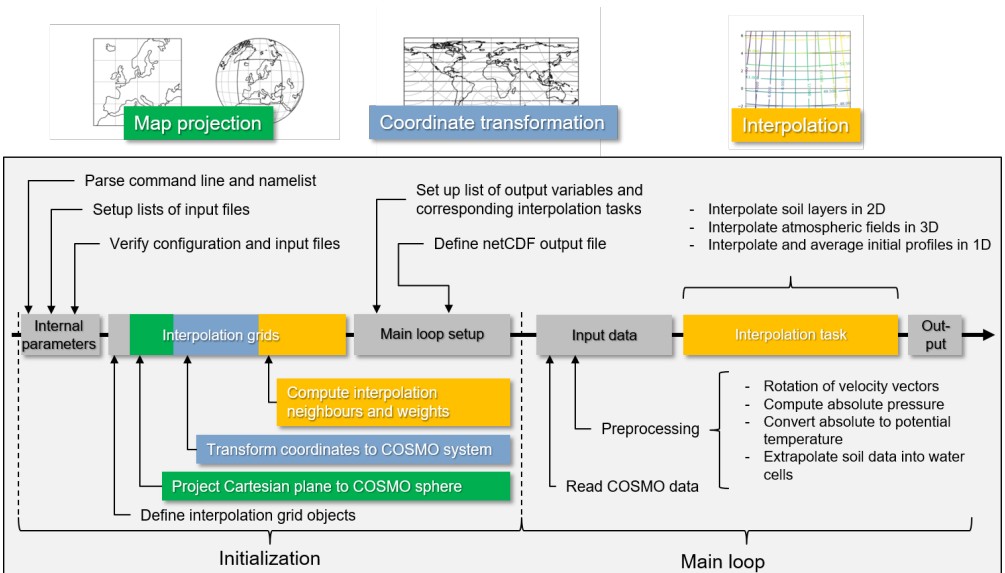

**Figure 6.** Diagram of INIFOR's program flow.

To obtain turbulent flow components $u_{i,\mathrm{b}}$ on the lateral boundaries, spatially and temporally correlated disturbances $u_i''$ are imposed onto the preliminary velocity components $\overline{u}_{i,\mathrm{b}}$,

$$u_{i,\mathrm{b}} = \overline{u}_{i,\mathrm{b}} + a_{ij}\, u_j'', \quad \text{with} \quad i,j \in 1,2,3\,. \tag{24}$$

$a$ is the amplitude tensor that is calculated from the Reynolds stress tensor $r$. To consider cross-correlations between the
5   velocity components, Lund et al. (1998) suggested a Cholesky decomposition to compute $a$ recursively by

$$a = \begin{pmatrix} \sqrt{r_{11}} & 0 & 0 \\ r_{21}/a_{11} & \sqrt{r_{22} - a_{21}^2} & 0 \\ r_{31}/a_{11} & (r_{32} - a_{21}\,a_{31})/a_{22} & \sqrt{r_{33} - a_{31}^2 - a_{32}^2} \end{pmatrix}. \tag{25}$$

Depending on characteristic length scales $L$ and time scales $T$ of the flow, which are defined individually for each velocity component in each spatial direction, $u_i''$ computes as

$$u_i''(t + \Delta t) = \Psi_i(t - \Delta t) \exp\left(-\frac{\pi \Delta t}{2T_i}\right) + \Psi_i(t)\left[1 - \exp\left(-\frac{\pi \Delta t}{T_i}\right)\right]^{0.5}, \tag{26}$$

10   with $\Delta t$ being the actual LES time step and the two-dimensional spatially correlated disturbances

$$\Psi_i^{m,l} = \sum_{j=-N_i}^{N_i} \sum_{k=-N_i}^{N_i} b_j\, b_k\, \zeta_i^{m+j,l+k}\,. \tag{27}$$

The subscripts $m$ and $l$ indicate grid positions at the lateral boundary, $N_i = 2L_i/\Delta x_i$, with $\Delta x_i$ being the grid spacing. $\zeta_i$ indicates a set of equally-distributed random velocities with zero mean and unit variance that are individually computed for


each velocity component. Finally, the spatial filter function computes as

$$
b_i = b_i^* \left( \sum_{k=-N_i}^{N_i} b_k^{*2} \right)^{-0.5},
\tag{28}
$$

with $b_i^* = \exp\left(-\frac{\pi|k|\Delta x_i}{L_i}\right)$. With this approach, the imposed $u_i''$ reflects the prescribed Reynolds stress as well as the spatial and temporal correlation according to $L_i$ and $T_i$, respectively.

From a mathematically point of view, the imposed fluctuations should have zero mean. Due to a finite sample of random numbers and the finite number of discrete grid points, however, the fluctuations have mean values slightly different from zero in practice. In order to overcome this, Kim et al. (2013) proposed a correction for the boundary normal flow component in order to maintain constant mass flux through the boundary. In order to avoid that the perturbations imposed onto the non-normal components may have non-zero mean too, e.g. non-zero mean $w-$ and $v-$component at the western model boundary, we correct the imposed turbulent velocity components as

$$
u_{i,\mathrm{corr}}'' = u_i'' - \frac{1}{S} \int_{\partial S} u_i'' dS,
\tag{29}
$$

with $S$ being the surface area of the respective lateral boundary.

For non-stationary flows, an inflow boundary can become an outflow boundary and vice versa. Hence, the turbulence generator is applied at each lateral boundary simultaneously, while at opposite boundaries (west and east, as well as north and south) we use the same $\Psi_i$ and thus the same set of random numbers (velocities). Further, perturbations are imposed at the end of each LES time step at the last Runge-Kutta substep right before the Poisson equation is solved to fulfill divergence-free flow.

From Eq. (27) it becomes obvious that the computational demand to calculate the spatially correlated disturbances is a function of the turbulent length scales – doubling the turbulent length scale leads to a quadrupling of the number of elements in the summation. For example, turbulent length scales vary significantly in time, reaching values $> 2000\,\mathrm{m}$ (see Fig. 13) and altering the computational demand of the turbulence generator within the course of the day. In order to balance the computational load over various processes, we parallelized the synthetic turbulence generator using the Message Passing Interface (MPI). To achieve this, we made use of the 2d domain decomposition used in PALM (Maronga et al., 2015). The imposed disturbances are computed locally on each MPI-process that belongs to a lateral boundary, so that no global communication is necessary. In order to avoid that only processes residing at the domain boundaries execute the computationally heavy code of Eqs. (27) and (28) while other processes are on hold, the computation of the filtered random numbers is divided in vertical direction by the number of processes in normal direction to the inflow boundary (e.g. on the left boundary, computation is distributed over npex parts where npex is the number of sub-domains, or MPI processes, along $x$), while the filtered random numbers are gathered on the boundary process, subsequently. In case of large $L_i/\Delta x_i$, this significantly reduces the required computational time of the synthetic turbulence generator (see Sect. 4.6). If $L_i/\Delta x_i$ is of only low value, however, the additional parallelization can be omitted by choice as the additional MPI communication can consume the benefit of the distributed calculation. Results from a performance test for the turbulence generator are discussed in Sect. 4.6.





The random numbers $\zeta$, which are defined on the discretized grid, are distributed over several MPI-processes and each process only knows its required set of random numbers. For example, suppose $\mathtt{nxl_A}$ ($\mathtt{nxl_B}$) and $\mathtt{nxr_A}$ ($\mathtt{nxr_B}$) are the left and right local domain boundary indices along the $x$-direction on MPI-process A (B), respectively, with $\mathtt{nxl_B} = \mathtt{nxr_A}+1$. On process A, random numbers need to be defined for the index range $\mathtt{nxl_A}-N_x : \mathtt{nxr_A}+N_x$; equivalently, on process B, $\mathtt{nxl_B}-N_x$
: $\mathtt{nxr_B}+N_x$, meaning that on process A and B random numbers partly overlap, e.g. within the index range $\mathtt{nxr_A}-N_x+1$ and $\mathtt{nxl_B}-N_x$ . In order to obtain the same $\zeta$ within the overlapping index range, we have to assure that the set of random numbers do not depend on the horizontal domain decomposition. This is achieved by linking the seed of the employed random-number generator to the grid index which is then independent on the domain decomposition. With respect to the computation of $\zeta$, MPI-communication can thus be reduced to only exchange data to compute its mean and variance.

In order to create time- and height dependent synthetic turbulence, respective information about the Reynolds stresses, as well as turbulent length and time scales for the velocity components are required. For stationary flows these information can be deduced from observations or from cyclic precursor simulations (Xie and Castro, 2008). However, for non-stationary flows with pronounced diurnal cycles and/or changing synoptic conditions running precursor simulations is practically not feasible. Also, to take these information from the mesoscale-model output is also not possible since these detailed information are often neither available nor part of the operational output. Hence, to allow for an adjustment of the synthetic inflow turbulence to changing atmospheric conditions, we parametrize the Reynolds stresses based on the time-dependent mesoscale inflow profiles. We follow the set of parametrizations presented by Rotach et al. (1996) which they employed in stochastic dispersion modelling. Please note, the following set of parametrizations refer to the stream- and spanwise components of the Reynolds stress that are not necessarily parallel to the $x$- or $y$-axis, respectively. In order to emphasize this, we indicate stream- and spanwise components with a tilde in the following.

Based on the original formulation by Brost et al. (1982), the variance of the streamwise flow component $\widetilde{r}_{11}$ is parametrized following Rotach et al. (1996)

$$\widetilde{r}_{11}(z) = u_*^2 \left( 0.35 \left( \frac{-z_i}{\kappa L_o} \right)^{2/3} + \left( 5 - 4\frac{z}{z_i} \right) \right), \quad \text{for} \quad z \leq z_i, \tag{30}$$

who added a correction term (first term in Eq. 30) proposed by Gryning et al. (1987) to account for unstable near-surface stratification. Here, $u_*$ is the friction velocity, $\kappa = 0.4$ the von-Kármán constant, $L_o$ the Obukhov length, $z_i$ the boundary-layer depth and $z$ being the height above ground. Similarly, we estimate the variance of the spanwise flow component by adding a correction term to the original formulation proposed by Brost et al. (1982):

$$\widetilde{r}_{22}(z) = u_*^2 \left( 0.35 \left( \frac{-z_i}{\kappa L_o} \right)^{2/3} + \left( 2 - \frac{z}{z_i} \right) \right), \quad \text{for} \quad z \leq z_i. \tag{31}$$

The profile of vertical velocity variance is taken from Gryning et al. (1987) as

$$\widetilde{r}_{33}(z) = w_*^2 \left( 1.5 \left( \frac{z}{z_i} \right)^{2/3} \exp\left[ -2\frac{z}{z_i} \right] + \left( 1.7 - \frac{z}{z_i} \right) \left( \frac{u_*}{w_*} \right)^2 \right), \quad \text{for} \quad z \leq z_i, \tag{32}$$



with $w_*$ being the convective velocity scale. The vertical transport of horizontal streamwise and spanwise momentum is estimated by Brost et al. (1982) as

$$\widetilde{r}_{31}(z) = -u_*^2 \left( \frac{z}{z_\mathrm{i}} - 1 \right), \quad \text{for} \quad z \leq z_\mathrm{i}, \tag{33}$$

and

$$\widetilde{r}_{32}(z) = -u_*^2 \left( 0.4 \frac{z}{z_\mathrm{i}} \left( 1 - \frac{z}{z_\mathrm{i}} \right) \right), \quad \text{for} \quad z \leq z_\mathrm{i}, \tag{34}$$

respectively. To our knowledge, there exist no comparable formulation to estimate $\widetilde{r}_{21}$ in the literature. Hence, we decided to simply set $\widetilde{r}_{21} = \sqrt{\widetilde{r}_{31}^2 + \widetilde{r}_{32}^2}$, assuming isotropy of horizontal and vertical transport of horizontal momentum. To estimate the boundary-layer depth for a wide range of stability regimes, including buoyancy- and purely shear-driven boundary layers, we calculated $z_\mathrm{i}$ from a bulk Richardson number criterion according to Heinze et al. (2017) based on the bulk Richardson number

$$Ri_\mathrm{b}(z) = \frac{g}{\theta_\mathrm{v,s}} \frac{\theta_\mathrm{v}(z) - \theta_\mathrm{v,s}}{u_\mathrm{h}(z)^2} \cdot z. \tag{35}$$

Starting at the surface, $z_\mathrm{i}$ is defined as the height where $Ri_\mathrm{b}$ first exceeds the critical bulk Richardson number $Ri_\mathrm{b,c} = 0.25$, which revealed to be a robust criterion to estimate the depth of the layer with significant turbulent transports caused by the presence of the surface (Heinze et al., 2017). Here, $u_\mathrm{h}$ denotes the horizontal wind speed from mesoscale model input, $\theta_\mathrm{v}$ the virtual potential temperature, $\theta_\mathrm{v,s}$ the virtual potential surface temperature inferred from the second prognostic level above the surface, following Heinze et al. (2017), and $g$ is the acceleration of gravity. In case of LOD = 1 input, $z_\mathrm{i}$ is determined based on the mean profiles prescribed at the lateral boundaries, while in case of LOD = 2 input ($xz-$ and $yz-$ slices of boundary data), $z_\mathrm{i}$ is determined locally at each $(x,y)$-boundary grid point and averaged horizontally afterwards.

The friction velocity $u_*$ used in Eqs. (30) to (32) is obtained by horizontally averaging of the friction velocity as used in the surface parametrization according to the Monin-Obukhov similarity theory (MOST) (see Maronga et al., 2020, Eq. 28) within the model domain. The same is done also to obtain $L_\mathrm{o}$. By doing this, we are aware that $u_*$ and $L_\mathrm{o}$, and thus also the parametrized Reynolds stress, are not entirely independent of each other, since adjustment effects of the turbulent flow may modify $u_*$ and $L_\mathrm{o}$ locally near the inflow boundaries, which in turn feeds back into the Reynolds stress parametrization again modifying $u_*$ and $L_\mathrm{o}$ near the inflow boundaries. However, for sufficiently large model domains, this feedback loop is negligible, as we discuss in Sect. 4.3. The convective velocity is computed as $w_* = (gH_0 z_\mathrm{i}/\theta_\mathrm{s})^{(1/3)}$, which is only defined for positive values of the mean surface sensible heat flux $H_0$, else it is set to zero so that $r_{3,3}$ remains defined. Even in neutral or stable boundary layers a vertical velocity variance can be observed. Hence, in order to account the Reynolds stress parametrization for a wide range of stability regimes, we followed Rotach et al. (1996) who replaced $w_*$ in Eq. 32 by the mixed velocity scale $w_\mathrm{m}$ (Troen and Mahrt, 1986) with

$$w_\mathrm{m} = \left( u_*^3 + \beta\, w_*^3 \right)^{\frac{1}{3}}, \tag{36}$$

with $\beta = 0.6$ according to Holtslag and Boville (1993).

Note that Eqs. (30), (31), (33) and (34) describe the flow characteristics in the streamwise and spanwise framework, which is indicated by the tilde. In a mesoscale nesting with changing wind directions, however, the streamwise and spanwise flow





directions do not necessarily coincide with the Cartesian grid axis which the prognostic velocity components relate to. Hence, the individual components of $\widetilde{r}$ are projected back onto the Cartesian grid by rotation about the vertical axis by the rotation angle defined by $\arctan(v/u)$.

Further, the synthetic turbulence generation requires information about the turbulent length and time scales. The turbulent
time scale of the flow is estimated according to Brost et al. (1982) with

$$T = 3.33\frac{z}{z_{\mathrm{i}}}\left(1 - 0.67\frac{z}{z_{\mathrm{i}}}\right). \tag{37}$$

Parametrizations of turbulent length scales exist for the lower part of the boundary layer, (e.g. Flay and Stevenson, 1988; Salesky et al., 2013; Li et al., 2017; Emes et al., 2019), however, no parametrization of turbulent length scales that cover the entire depth of the boundary layer and all stability regimes exists to date to our knowledge. Hence, we calculate turbulent
length scales of the flow components according to Tennekes and Lumley (1972) using the parametrized Reynolds stress and the timescale:

$$L_i = T\sqrt{r_{ii}}, \tag{38}$$

with $i \in 1, 2, 3$ indicating the streamwise, spanwise and vertical direction, respectively.

Assuming that turbulence is only present within the boundary layer, $r$, $T$, and $L$ are faded for $z > z_{\mathrm{i}}$ with

$$\Phi(z) = \Phi(z_{\mathrm{i}}) * \exp\left[\frac{-9.3}{L(z_{\mathrm{i}})}(z - z_{\mathrm{i}})\right], \quad \text{with} \quad \Phi \in \{r, T, L\} \quad \text{and} \quad z > z_{\mathrm{i}}. \tag{39}$$

Here, the fading function is designed so that $\Phi(z)$ rapidly decreases above the boundary layer.

In case of non-stationary flows, the turbulence parameters $r$, $T$, and $L$ are adjusted hourly by default, but the frequency can be also modified by the user. We note that updating the turbulence parameters violates the temporal correlation expressed in Eq. (26). Hence, we performed a test simulation where we omitted the first term in Eq. (26) after updating the turbulence pa-
rameters and we found no difference with respect to the spatial development of the flow (not shown), indicating that occasional violations of the temporal correlation have no significant effect on the development of the flow.

## 3  Simulation setup and statistical analysis

In order to test the implemented mesoscale nesting approach, we selected a particular weather scenario with a developing daytime convective boundary layer (CBL) that features advective conditions with moderate wind speeds and changing mean-
wind direction. Moreover, the scenario is characterized by little to no cloud cover which is attributed to the fact that PALM cannot capture high-altitude clouds yet (due to missing ice-phase physics, planned) and thus cannot realistically reproduce the prevailing radiative forcing. We simulated one diurnal cycle of the evolving CBL starting at 0 UTC on May 7$^{\text{th}}$, 2016, for a domain located east of Berlin, Germany. The boundary layer on this day was characterized by clear-sky conditions and moderate mean boundary-layer wind speeds of about $7 - 8\,\mathrm{m\,s^{-1}}$ from the north-east, later turning to the south-east during the
morning hours. Figure 7 shows horizontal mean vertical profiles of horizontal wind $u_{\mathrm{h}}$, potential temperature $\theta$ and specific



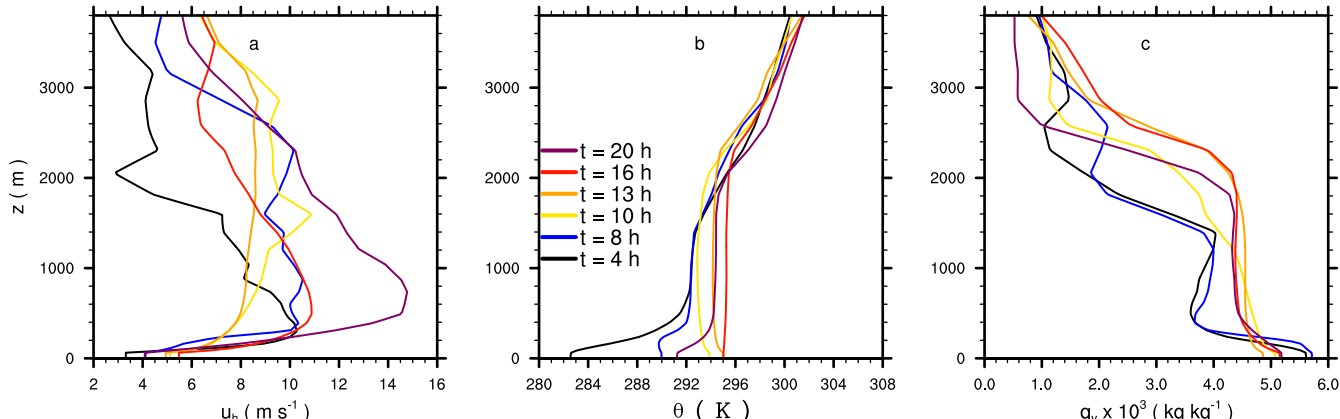

**Figure 7.** COSMO-derived vertical profiles of **(a)** horizontal wind speed, **(b)** potential temperature, and **(c)** specific humidity prescribed at the lateral domain boundaries at different points in time after simulation start at 0 UTC.

humidity $q_v$ at different points in time obtained from COSMO. During nighttime the profiles indicate a stably-stratified layer up to $z = 800\,\mathrm{m}$, while at 08 UTC the stably stratified layer gets successively eroded by the beginning surface heating. Later in the day, a well-mixed CBL develops with maximum $z_\mathrm{i} \approx 2400\,\mathrm{m}$. At about 16 UTC, the evening transition sets in and again a stably stratified layer develops near the surface.

We assumed a horizontally homogeneous and flat surface instead of the particular terrain, land use, and buildings at the Berlin area. We made this idealization in order to be able to determine adjustment fetch lengths under time-dependent inflow conditions. Surface heterogeneities in terms of land use or terrain would modify the turbulent flow field locally, making it impossible to disentangle changes in the turbulent flow due to adjustment effects and surface heterogeneity. We employed the embedded land-surface model (see Maronga et al., 2020; Gehrke et al., 2020) to obtain sensible and latent heat fluxes at
the surface, which we assumed to be fully covered with short grass. We chose this setup as a trade-off roughly reflecting the prevailing land use in this area with distributed farm- and grassland, forest patches and urbanized environments.The soil was initialized with horizontally homogeneous profiles of soil temperature and soil moisture taken from COSMO. Since the soil properties in COSMO are aggregated over various surfaces and soil types, the soil conditions are not necessarily in equilibrium with the assumed grass surfaces as well as selected atmospheric conditions. Hence, we ran a two-day surface spinup as a
precursor to the 3D simulation (Maronga et al., 2020) in order to bring the soil into equilibrium with the atmospheric conditions and to avoid spinup effects that may lead to varying heat fluxes at the beginning of the simulation. The incoming short- and longwave radiation was modelled using the Rapid Radiative Transfer Model for Global Models (RRTMG, Clough et al., 2005).

The simulated domain is located at 52.5°N, 13.4°E (PALM origin) and extends over $43.2 \times 43.2 \times 4.7\,\mathrm{km}^3$ in the $x$-, $y$-, and $z$-direction, respectively, with an isotropic grid spacing of $25\,\mathrm{m}$. Above $z = 3\,\mathrm{km}$ – approximately $600\,\mathrm{m}$ above maximum
boundary layer depth – the vertical grid was successively stretched up to 50 m vertical grid spacing. This allows for proper resolution of the convective boundary layer, though we note that the nighttime stably-stratified boundary layer is only poorly





represented with the chosen grid spacing. The advection terms were discretized with a fifth-order upwind scheme (Wicker and Skamarock, 2002); for the time-stepping we applied a third-order Runge-Kutta method according to Williamson (1980).

We performed two simulations with different lateral boundary conditions. A first simulation, hereafter referred to as REF, where the boundary conditions were given as LOD = 1, i.e. horizontally-averaged profiles. Unless not further noted, we refer

to this simulation in the following analysis. And a second simulation, where heterogeneous boundary values were prescribed (LOD = 2). This second simulation will be used to check whether the LES simulation results depend on grey-zone related roll convection in COSMO.

In order to evaluate the spatial development under non-stationary atmospheric conditions where the flow direction is mostly not aligned with the horizontal grid axis, phase-averaging of turbulence statistics along one grid axis is not possible. Hence,

we calculated resolved-scale variances of the velocity components by

$$\langle u_i' u_i' \rangle = \langle u_i u_i \rangle - \langle u_i \rangle \langle u_i \rangle \,, \quad \text{with} \quad i \in (1,2,3) \,, \tag{40}$$

while the angle brackets indicates a time average over half an hour and the prime indicates the turbulent fluctuation. The resolved-scale TKE was computed as $TKE = 0.5 \cdot \sum \langle u_i' u_i' \rangle$. For each grid point location we determined its distance to the inflow boundary at the given wind direction. Therefore, we calculated virtual backward trajectories for each half-hour interval

from the current mean wind direction; subsequently, we determined the distance $d$ between the sampling location and the intersection point of the backward trajectory with the closest inflow boundary. Finally, variances were averaged over similar distances to the inflow boundary, while we sorted similar distances into equally-sized bins of 100 m to obtain sufficiently large sample size for each discrete distance.

Please note, in this study we will mainly focus on convective conditions, especially with respect to the spatial development

of the flow. The nighttime stable flow is only poorly resolved at the given grid spacing, making it difficult to make reliable conclusions concerning the flow adjustment. Here we will refer to follow-up studies.

## 4   Results

In the following section we show results from a mesoscale nested LES for a diurnal cycle. In order to better guide the reader through this section, we will first give a short outline of what to expect: First, we describe the boundary-layer structure and

its development over the diurnal cycle in the LES as well as in COSMO. In the subsequent, we discuss differences between the LES and COSMO with respect to the boundary-layer representation and its implications in a nested simulation. In the following, triggered by imposed time-dependent synthetic turbulence, we focus on the spatial development of the turbulent flow within the LES domain and determine adjustment lengths where the turbulent flow is fully developed. In addition, we present results on how roll-like structures emerging in the COSMO simulation propagate into the LES. Moreover, we discuss

implications near the LES domain inflow and outflow boundary. Finally, we look at a more technical issue and demonstrate the computational efficiency of the synthetic turbulence generation.



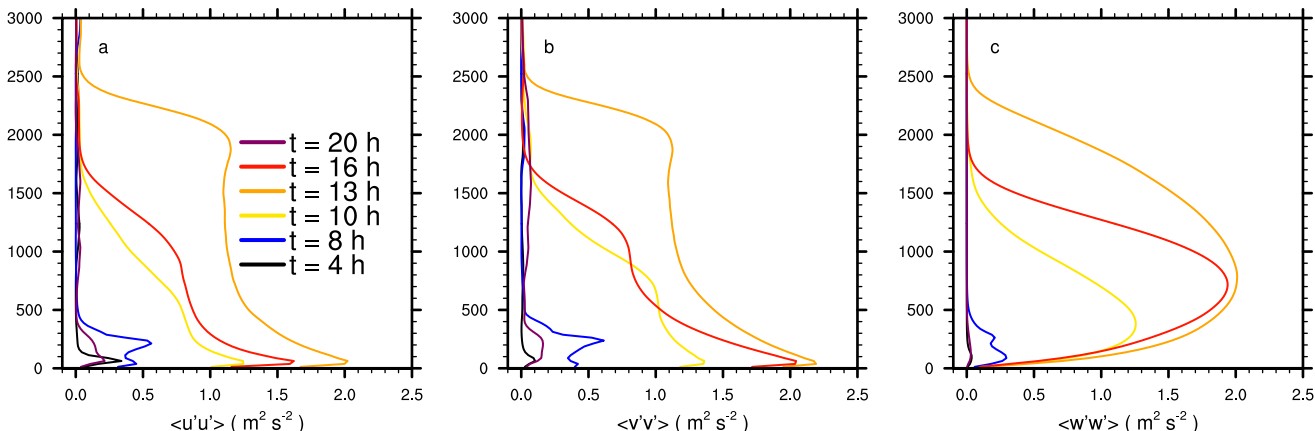

**Figure 8.** Variances of the velocity components: **(a)** for the $u$-component, **(b)** for the $v$-component and **(c)** for the $w$-component. The variance are computed from the region where the turbulent flow has been already adjusted.

## 4.1 Boundary-layer structure

Figure 8 shows vertical profiles of the velocity variances horizontally averaged over a $10 \times 10\,\mathrm{km}^2$ at the center of the domain where the turbulent flow has been already adjusted (see Sect. 4.3). The variances show a pronounced diurnal cycle, with only small values during nighttime and the morning hours. At 8 UTC, when the surface heating sets in, a double-peaked profile can be observed. Later, the variances increase, with the horizontal variances exhibiting a maximum near the surface. With increasing height the horizontal variances decrease, while at 13 UTC a secondary peak can be observed near the boundary-layer top. The vertical variances peak in the middle part of the boundary layer and approach zero at boundary-layer top. In the afternoon and evening hours the value of the variances again decreases and the boundary becomes shallower. Overall, the variances show a typical diurnal cycle for clear-sky conditions (André et al., 1978).

Figure 9 shows vertical profiles of the potential temperature at the inflow boundary and from the inner part of the domain, where the turbulent flow is spatially fully developed. At 09 UTC (dashed lines), the COSMO inflow profile indicates a warmer boundary layer within the lowest $500\,\mathrm{m}$ compared to the inner-domain profile, while further above (within the residual layer) the profiles from COSMO and the inner domain coincide. At 10 UTC, the imposed COSMO inflow profile already indicates an unstable stratification within the lower part of the boundary layer and a well mixed layer up to $z = 2000\,\mathrm{m}$, while the inner-domain profile indicates an unstable stratification only within the surface layer and a well-mixed layer above reaching only up to $z = 1400\,\mathrm{m}$ where the potential temperature profile indicates similar values compared to the profile one hour before. This means that between 09–10 UTC the boundary layer in COSMO develops more rapidly, where the stably stratified layer gets completely eroded, the residual layer becomes convective and the boundary layer grows significantly, while in PALM the boundary layer develops less rapidly and only the stably stratified layer gets eroded. At 13 UTC, the shapes of the COSMO inflow and inner-domain profiles are similar, indicating similar boundary-layer depth, though the inflow profile indicates a



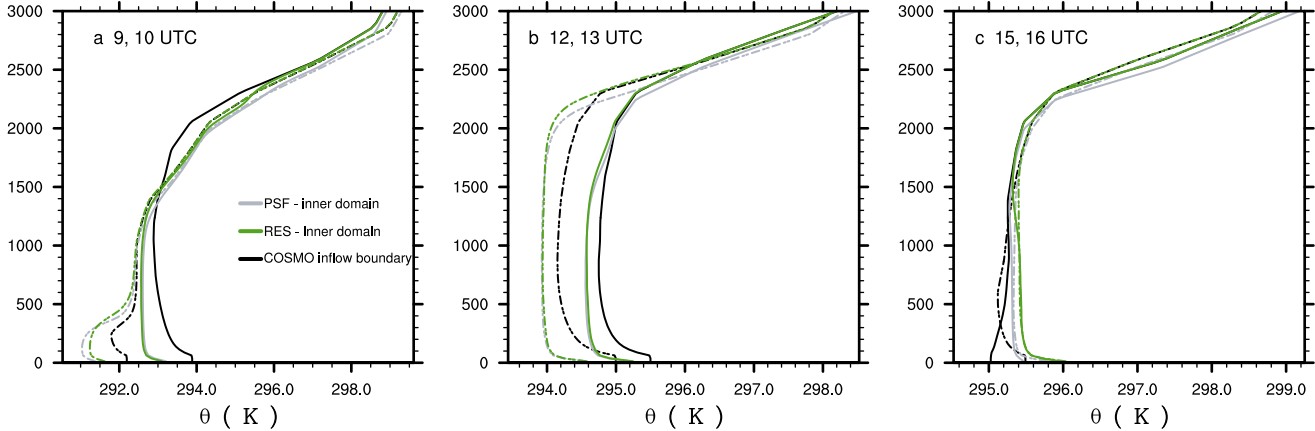

**Figure 9.** Vertical profiles of potential temperature prescribed at the inflow boundary as well as within the inner part of the domain where the flow has been spatially fully developed. Profiles are shown for simulation RES (green) as well as for a test simulation PSF (grey) with prescribed surface heat fluxes obtain from COSMO. Profiles are shown for **(a)** 9 and 10 UTC, **(b)** 12 and 13 UTC, and **(c)** 15 and 16 UTC as dashed and solid lines, respectively.

warmer boundary layer compared to the region farther downwind of about $0.25\,\mathrm{K}$. At 16 UTC, the COSMO inflow profile indicates already a weakly stable stratification below $z = 900\,\mathrm{m}$, while the inner-domain profile still shows a vertically well-mixed boundary layer. At all points in time shown, the potential temperature profiles above the boundary layer do not change significantly, indicating only small horizontal temperature advection on the mesoscale. Indeed, we calculated the advection

tendency at 10 UTC and 16 UTC to $-0.1\,\mathrm{K\,h^{-1}}$ and $0.05\,\mathrm{K\,h^{-1}}$ (not shown), respectively, meaning that horizontal advection of temperature cannot explain the mismatch of the temporal boundary-layer development between COSMO and PALM.

Figure 10 shows the surface net radiation and surface heat fluxes for COSMO and simulation REF during the course of the day. During the night, the surface net radiation and sensible heat flux are significantly smaller in COSMO, indicating more cooling of the surface which results also in more negative surface sensible heat flux. At daytime, COSMO and PALM show a

similar diurnal cycle of surface net radiation with comparable peak values at noon and only small differences during the course of the day. The available energy at the surface, however, is differently partitioned into the surface latent and sensible heat flux between PALM and COSMO. The sensible heat flux in COSMO shows slightly higher values compared to PALM between 10 UTC and 12 UTC, but significantly lower values during the afternoon hours where the sensible heat flux approaches zero at about 16 UTC corresponding to the stabilization of the COSMO-simulated boundary layer (see Fig. 7 and 9), whereas PALM

still simulates a positive sensible heat flux of $> 100\,\mathrm{W\,m^2}$. In contrast, the surface latent heat flux in COSMO shows larger values in the afternoon compared to PALM, meaning that the bulk of the available energy is partitioned into the surface latent heat flux being not available for heating the boundary layer.

In addition, we performed a simulation where the diurnal cycle of $H_0$ and $LE_0$ was explicitly prescribed, hereafter called simulation PSF, rather than computed by the land-surface model. This test was motivated to check whether the discrepancy





between the inner-domain and the COSMO inflow profile is attributed to a possible misrepresentation of the surface energy balance attributed to the idealized setup with a homogeneous grass surface rather than a more realistic surface. However, except for minor differences with a slightly cooler boundary layer in the morning (see Fig. 9), a slightly shallower boundary layer at noon, and a slightly cooler boundary layer in the afternoon, we found no significantly different structure of the boundary layer

between simulation PSF and RES. This shows that the discrepancy between the inner-domain and the COSMO inflow profile can not be attributed to any misrepresentation in the surface energy balance compared to COSMO but to some other reason.

As the COSMO profiles are mapped onto the inflow boundary, the more rapid evolution or the earlier stabilization of the boundary layer at 10 UTC and 16 UTC, respectively, also propagate into the PALM model domain. Figure 11 shows vertical cross-sections of the potential temperature and corresponding boundary-layer height. At 10 UTC and 13 UTC, according to the

higher potential temperature at the inflow boundary shown in Fig. 9, the potential temperature within the boundary layer and the boundary-layer height decrease with increasing distance to the inflow boundary, which indicates that at these points in time, a deeper and warmer boundary layer is advected into the PALM domain. In contrast, at 16 UTC where the inflow potential temperature profile (see Fig. 9) already indicates a stable inflow with lower values of potential temperature, the potential temperature and boundary-layer height increase with increasing distance to the inflow boundary (Fig. 11c,d). Especially the

spatial gradient of the boundary-layer height close to the inflow boundary indicates that a significantly shallower boundary layer is advected into the PALM domain which further propagates downwind (Fig. 11d) later on. Also, with increasing distance to the inflow boundary, more and deeper convective updrafts, indicated by higher values of potential temperature, can be observed, while close to the inflow boundary only shallow convective updrafts occur. This suggests that the stable stratification near the inflow boundary suppresses convection in the later afternoon. In particular the horizontal difference in the boundary-

layer structure at 16 and 16:30 UTC shows that temporal changes on the inflow temperature (and humidity, not shown) reach the inner part of the model domain with a time-lag. In this setup, it takes about 1 to 1.5 hours until the signals imposed at the inflow boundary reach the outflow boundary, meaning that the boundary layer becomes horizontally heterogeneous during the transition phase. We note that this is in contrast to the large-scale forcing approach by Heinze et al. (2017) where large-scale advection and nudging terms were considered at each location at the same time so that the LES solution can approach the

mesoscale solution as a whole, though also with this approach the transition of the LES towards the mesoscale mean state is time-lagged according to the applied nudging time scale.

The temporal change in inflow conditions can also be observed visually in the vertical velocity shown in Fig. 12. At 10 UTC, where a deeper boundary layer accompanied with more energetic synthetic disturbances is advected into the model domain, the up- and downdrafts close to inflow boundary show a larger amplitude compared to the up- and downdrafts further downwind.

In contrast, at 16 UTC, where a more shallow and stable boundary layer accompanied with only small synthetic disturbances (see Fig. 13) is advected into the model domain, the amplitude of the up- and downdrafts is only small near the inflow boundary and increases farther downwind.

In summary, the boundary-layer evolution in the LES and the mesoscale model is not fully synchronous, especially in the morning and afternoon hours. As the LES obtains its boundary conditions from the mesoscale solution, the changes implied by

the inflow conditions propagate through the LES domain, independent of whether differences emerge from changing synoptic



**Figure 10.** Time series of surface net radiation (top) and surface latent and sensible heat flux (bottom) from COSMO (dashed) and PALM (solid). Values are horizontally averaged over the corresponding domain. The fine dashed horizontal line indicates zero surface net radiation and heat fluxes, respectively. Please note the different temporal resolution between PALM and COSMO, with COSMO values are only defined hourly.




**Figure 11.** Instantaneous $x$-$z$-cross-section of the simulated potential temperature (contours) and local boundary-layer height (black solid line) in PALM at **(a)** 10 UTC, **(b)** 13 UTC, **(c)** 16 UTC, and **(d)** 16:30 UTC. The boundary-layer height is calculated according to the Richardson bulk criterion. The inflow boundary is on the right. Please note, the inflow direction is from the south-east, meaning that the $x$-axis does not correspond with the distance to the inflow boundary. The cross-section is taken at $y = 32200\,\mathrm{m}$ where the potential temperature is not affected by advection from the southern inflow boundary. Please note the changing contour-levels between the different points in time.



**Figure 12.** Horizontal cross-sections of the instantaneous vertical velocity component at $z = 500\,\mathrm{m}$ for **(a)** 10 UTC, **(b)** 13 UTC, and **(c)** 16 UTC, with mean wind blowing from the south-east.



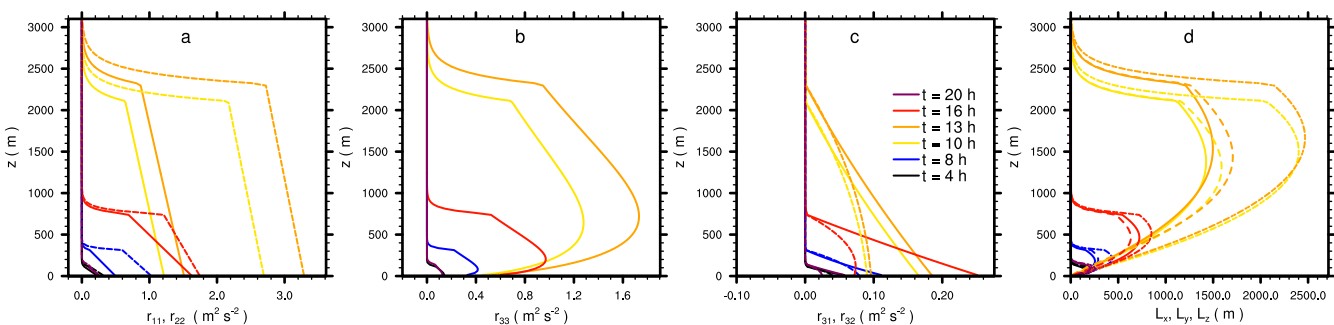

**Figure 13.** Parametrized components of the Reynolds stress tensor as well as turbulent length scales at different points in time. In **(a)** and **(c)** the solid (dashed) line belong to $r_{11}$ ($r_{22}$) and $r_{31}$ ($r_{32}$), respectively. In **(d)** the solid, fine and coarse dashed line belong to $L_x$, $L_y$ and $L_z$, respectively.

conditions or from discrepancies in the boundary-layer representation between the LES and the mesoscale model. Thus, the LES solution is nudged towards the mesoscale solution any way, meaning that possible model biases (of the mesoscale model) also propagate into the LES.

## 4.2 Characteristics of imposed turbulence

The parametrized Reynolds-stress components depend on the inflow profiles obtained from the mesoscale model input, i.e. $z_i$, as well as on $u_*$, $w_*$ and $H_0$ obtained from the inner part of the model domain, which determines the depth and the amplitude of the Reynolds-stress components, respectively.

Figure 13 shows the characteristics of the synthetic turbulence imposed at the lateral boundaries. During nighttime, Reynolds stresses and turbulent length scales are only small and defined in a shallow layer up to about $z = 200\,\mathrm{m}$, attributed to the shallow

boundary-layer depth of the stably stratified layer. Later, when convection sets in and the boundary-layer depth increases, the values of the Reynolds-stress components and the length scales increase as well, with maximum length scales of about $L = 2500\,\mathrm{m}$ at 13 UTC. The shape of the Reynolds-stress and length-scale profiles does only change slightly during the simulation due to the turning wind direction where the contribution from the streamwise- and the spanwise parametrization to the components on the Cartesian grid slightly changes. Above the boundary-layer top, the Reynolds-stress components and

length scales decrease rapidly to zero with height. For instance, at 13 UTC at about $z = 2700\,\mathrm{m}$ the length scales approach zero, while the amplitude of the imposed disturbances are already nearly zero, meaning that only small perturbations are added above the boundary-layer top. Qualitatively, the parametrized Reynolds-stress components resemble the variance profiles shown in Fig. 8, though $r_{11}$ ($r_{22}$) are under-(over)estimated compared to $u'u'$ ($v'v'$) around noon, respectively, and do not account for the secondary peak near the boundary-layer top. Furthermore, at 10 UTC (and 16 UTC) it strikes that the Reynolds-

stress components indicate a deeper (shallower) boundary layer compared to the velocity variances, respectively, which is in





accordance with the horizontally heterogeneous structure of the boundary layer during transition periods (see Sect. 4.1), which again is owed to the fact that the onset and offset of convection is shifted between COSMO and PALM.

In summary, the parametrized Reynolds-stresses resemble the variances profiles created by the LES itself reasonably well during the course of the day. However, we emphasize that the imposed turbulence is only considered to be a rough description

to resemble the second-order statistics of a fully adjusted flow, while its spectral distribution, or higher-order moments are not accounted for.

### 4.3    Spatial development of the flow

Figure 14 shows the spatial development of the resolved-scale TKE depending on the distance to the inflow boundary at different heights and points in time. Please note, the upper abscissa depicts the dimensionless distance to the inflow boundary using

convective scaling, which was originally developed by Willis and Deardorff (1976) to scale Lagrangian dispersion experiments. Here, we use this to scale the travel-time $d/u_h$ of a signal imposed at the lateral boundary with the eddy-turnover time $z_i/w_*$ in the CBL, where $d$ indicates the distance to the inflow boundary in wind direction and $u_h$ indicates the mean boundary-layer wind speed (averaged over the depth of the boundary layer). At 10 UTC, the TKE peaks at about 3 km and 5 km downstream of the inflow boundary within the lower and upper part of the CBL, respectively. This can also be observed visually in Fig. 12

where the amplitude of the up- and downdrafts close to the inflow boundary appears stronger. Farther downstream, the TKE gradually decreases, approaching an equilibrium value at about 10 km downstream of the inflow boundary. At 13 UTC the TKE peaks again close to the inflow boundary, though the amplitude of the peak value is smaller and closer to it equilibrium value where the flow has been spatially fully developed. Especially within the upper part of the CBL, the TKE reaches a nearly constant value at about 6 km downstream of the inflow boundary. The dimensionless distance until the flow has been devel-

oped is about $1.5\,u_h z_i/w_*$ at 10 UTC and 13 UTC, meaning that the flow needs more than one eddy turnover to become fully developed.

At 16 UTC the situation becomes qualitatively different. Even though the TKE within the lower part of the CBL peaks again close to the inflow boundary and approaches a nearly constant value farther downstream, it gradually increases starting at about 20 km downstream. Within the middle and upper part of the CBL, the TKE is close to zero near the inflow boundary,

according to the only small amplitude of the imposed synthetic turbulence (see Fig. 13), and gradually increases towards the outflow boundary. This can also be observed in Fig. 12 where the up- and downdrafts near the inflow boundary are only weak and become stronger further downstream, indicating that turbulence first needs to develop spatially. Due to the horizontally heterogeneous boundary layer with already weakly stable boundary conditions near the inflow boundary and still convective conditions farther downstream, the TKE does not reach an equilibrium value and a spatial adjustment length cannot be

accurately determined for this point in time.

In order to investigate how the structure of the turbulent flow develops, Fig. 15 shows the corresponding horizontal profiles for the skewness of the vertical velocity component. As typically observed in a clear-sky convective boundary layer, the skewness is positive and increases with height (Moeng and Rotunno, 1990). Near the inflow boundary the skewness is close to zero or even slightly negative indicating that the imposed up- and downdrafts are equally distributed with respect to their area





**Figure 14.** Horizontal profiles of 30-min time-averaged resolved-scale TKE depending on the distance to the inflow boundary. The TKE is shown for the lower-, middle- and upper part of the boundary layer at **(a)** 10 UTC, **(b)** 13 UTC, and **(c)** 16 UTC. The lower abscissa indicates the absolute distance to the inflow boundary, while the upper abscissa indicates the travel time of an imposed signal normalized with the eddy-turnover time. The TKE is averaged over similar distances to the inflow boundary.





**Figure 15.** Horizontal profiles of 30-min time-averaged skewness $S_w$ of $w$ depending on the distance to the inflow boundary. $S_w$ is shown for the lower-, middle- and upper part of the boundary layer at **(a)** 10 UTC, **(b)** 13 UTC, and **(c)** 16 UTC. The lower abscissa indicates the absolute distance to the inflow boundary, while the upper abscissa indicates the travel time of an imposed signal normalized with the eddy-turnover time. $S_w$ is averaged over similar distances to the inflow boundary.



contribution. Similar to the TKE, the skewness peaks close to the inflow boundary and adjusts towards a constant positive value farther downstream where strong/narrow thermal updrafts and weaker/wider downdrafts are in equilibrium. This equilibrium value is reached earlier for the skewness compared to the TKE (see Fig. 14), which is in accordance to the results shown in Muñoz-Esparza and Kosović (2018). In other words, the flow rapidly develops coherent turbulent structures that are in

equilibrium, albeit these are still too energetic as indicated by the TKE.

    To investigate of how fast land-atmosphere interactions adjust, Figure 16 shows horizontal profiles of $LE_0$, $H_0$ and $u_*$ depending on the distance to the inflow boundary. At 10 and 13 UTC, right behind the inflow boundary, $H_0$ and $u_*$ increase with increasing distance and approach a nearly constant value after about 1.5–3 km, indicating almost homogeneous fluxes of sensible heat and horizontal momentum. Likewise, $LE_0$ approaches a nearly constant value after about 1.5–3 km, though,

in contrast to $H_0$ and $u_*$, it decreases slightly behind the inflow boundary. At 16 UTC the fluxes behave similar though it takes a slightly longer distance of about 5–6 km to approach a nearly constant value. Compared to the TKE, the land-atmosphere exchange adjusts faster and does not show a significant dependence on the distance to the inflow boundary, e.g. at 16 UTC surface fluxes are homogeneous after they have been adjusted, while the TKE continues to gradually increase farther downstream. Especially the fact that the fluxes rapidly approach a homogeneous value is important for the parametrization of

the Reynolds-stress components. The surface fluxes directly enter the parametrization, see Eqs. (30)–(34) and (36), so that the imposed synthetic turbulence depends on the domain-averaged surface fluxes including the region near the inflow boundary. However, since the fluxes rapidly approach a homogeneous value, the error made by averaging is not significant for sufficiently large model domains.

    Finally, we would like to note that we also simulated different scenarios with higher (2UH) and lower (05UH) wind speeds,

as well as with increased (15RS) and reduced (075RS) shortwave solar radiation, in order to test the convective scaling. Even though the peak amplitudes in the horizontal TKE profile are different due to the modified forcing (see Fig. 17), the peak locations coincide with respect to $dw_*/(u_h z_i)$ at the respective height levels. This, in turn, indicates that the required distance needed to allow the flow to spatially develop is not an universal number but scales with $u_h z_i/w_*$ in a convective boundary layer, meaning that with higher wind speeds or less surface heating the required adjustment fetch increases (see Tab. 4).

Summarized, under convective conditions the turbulent flow is fully developed within the boundary layer after about $1.5 u_h z_i/w_*$ and further adjustment effects farther downstream are only small. The adjustment lengths we observe in our mesoscale nesting approach are comparable to those observed in mesoscale WRF-LES nesting with the cell perturbation method applied (Muñoz-Esparza et al., 2015; Muñoz-Esparza and Kosović, 2018). However, we note that the absolute distance required to allow for fully spatially developed turbulence is still in the order of kilometers, meaning that the model domain

should be sufficiently large to place the region of interest sufficiently apart from the inflow boundaries.

### 4.4 Effect of heterogeneous inflow conditions

Figure 18c shows a horizontal cross-section of the vertical velocity component from COSMO within the middle part of the boundary layer at 12:00 UTC. The COSMO simulation shows elongated structures that are mainly orientated along the mean-wind direction with up- and downdrafts in the order of $\mathrm{ms}^{-1}$. The wavelength of these structures is $\approx 0.05 - 0.1°$, which



**Figure 16.** Horizontal profiles of 30-min surface latent $LE_0$ and sensible $H_0$ heat flux as well as friction velocity $u_*$ depending on the distance to the inflow boundary at **(a)** 10 UTC, **(b)** 13 UTC, and **(c)** 16 UTC. The lower abscissa indicates the absolute distance to the inflow boundary, while the upper abscissa indicates the travel time of an imposed signal normalized with the eddy-turnover time. $H_0$ and $LE_0$ are plotted with respect to the left ordinate, while $u_*$ is plotted with respect to the right ordinate. The shown values are averaged over similar distances to the inflow boundary.





**Figure 17.** Horizontal profile of 30-min time-averaged resolved-scale TKE on 13 UTC depending on the normalized distance to the inflow boundary at **(a)** $0.1\,z_i$, **(b)** $0.5\,z_i$, and **(c)** $0.75\,z_i$.





**Figure 18.** Horizontal cross-sections of the instantaneous vertical velocity component $(\mathrm{m\,s^{-1}})$ at 12 UTC at $z = 500\,\mathrm{m}$ from **(a)** a PALM simulation with heterogeneous lateral and top boundary values given (LOD=2), **(b)** a PALM simulation with homogeneous boundary values given (LOD=1) and **(c)** from a COSMO simulation. The black box in **(c)** indicates the location of the nested PALM domain within COSMO. The horizontal grid spacing in COSMO is $0.025\,°$. Please note, for sake of comparison the PALM axis are plotted in $(°)$ as well, even though PALM uses Cartesian coordinates.





**Table 4.** Scaling parameters at 13 UTC for the sensitivity simulations.

| Case | $z_i\,(\mathrm{m})$ | $w_*\,(\mathrm{m\,s^{-1}})$ | $u_h\,(\mathrm{m\,s^{-1}})$ | $1.5 \cdot \frac{u_h z_i}{w_*}\,(\mathrm{km})$ |
|------|------|------|------|------|
| REF | 2510.0 | 2.22 | 8.0 | 13.5 |
| 2UH | 2540.0 | 1.84 | 14.7 | 20.0 |
| 05UH | 2450.0 | 2.05 | 3.5 | 4.5 |
| 15RS | 2510.0 | 2.82 | 8.0 | 7.0 |
| 075RS | 2510.0 | 1.82 | 8.0 | 11.0 |

is in the order of COSMO's horizontal grid spacing of $0.025°$ ( 2.8 km). Previous studies with the WRF model revealed that these kind of up- and downdrafts are a numerical artefact rather than a realistic feature of the boundary layer when the boundary layer depth is within the range of the horizontal grid spacing (Ching et al., 2014; Zhou et al., 2014; Shin and Dudhia, 2016). Even though results from mesoscale WRF simulations are not necessarily transferable one to one to COSMO,

we nevertheless assume a similar behaviour here. With a boundary-layer depth of $\approx 2.4$ km at 12:00 UTC (see Fig. 7), the dominant length scales of the flow approach the horizontal grid spacing, meaning that convection can partly be resolved on the COSMO grid. Figure 18a,b show corresponding horizontal cross-sections of the vertical velocity component from a PALM simulation with heterogeneous and homogeneous boundary values prescribed, respectively. After some adjustment behind the inflow boundaries (east and south boundary) where turbulent structures are weaker and appear on smaller scales (see Sect.

4.3), elongated structures orientated along the mean wind direction form in both simulations, with typical strength of up- and downdrafts for a convective boundary layer. In the heterogeneous case, however, the elongated turbulent up- and downdrafts appear more clustered with similar wavelength as in the COSMO simulation, while in the homogeneous case the turbulent up- and downdrafts are more homogeneously distributed. These clustering of up- and downdraft in the PALM simulation indicates that grid-dependent flow structures resolved by COSMO propagate into the LES domain and trigger the development

of elongated structures in PALM with similar wavelength that persists throughout the entire model domain. This is in contrast to Mazzaro et al. (2017), who found that with the cell-perturbation method grid-dependent structures do not significantly bias the turbulence behind the turbulence-adjustment region, though they might affect the rate of evolution of turbulence near the inflow boundaries. In this setup these under-resolved roll-like structures only occurred for 2 hours around noon and drifted slightly so that the elongated rolls in the LES domain were non-stationary (not shown). However, in case of stationary rolls, these grid-

dependent structures of the mesoscale model may introduce a location bias to the turbulent flow, so that the PALM solution becomes dependent on the presence of unrealistic flow structures in the mesoscale model. For convective conditions with under-resolved mesoscale convection, we thus recommend to use homogeneous boundary conditions rather than heterogeneous ones, though we are aware that especially for large LES model domains gradients on the mesoscale may vanish.

### 4.5 Implications near the inflow and outflow boundaries

In this section, we focus on the flow near the inflow and outflow boundaries. Due to different model representations of surface processes and different surface input data, the mesoscale near-surface wind and temperature profiles can deviate from the one



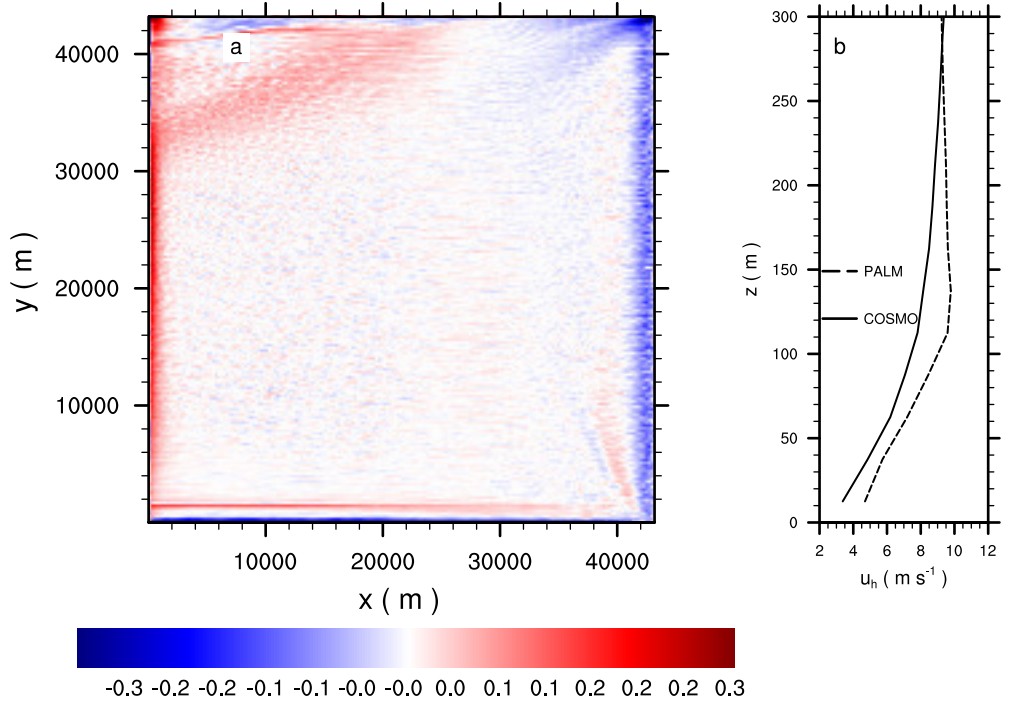

**Figure 19.** Horizontal cross-section of 30-minute time-averaged vertical velocity component at 03 UTC at $z = 100\,\mathrm{m}$ **(a)**. In addition, **(b)** shows corresponding vertical profiles of the horizontal wind speed averaged over a $10 \times 10\,\mathrm{km}$ area located at the domain center, as well as the lateral inflow profile from the COSMO solution. The wind blows from the east.

the LES would simulate. As an example, Fig 19b shows the near-surface wind profiles taken at the inflow boundary (COSMO solution) and taken from the inner part of the LES model domain about 20 km downstream of the inflow boundary. Within the lower $200\,\mathrm{m}$ the PALM profile shows a higher wind speed compared COSMO, indicating that the imposed inflow profile is not in equilibrium with the surface in PALM. As a consequence, the horizontal near-surface flow is accelerated behind the inflow

5 boundary (east and south boundary), meaning that horizontal momentum needs to be transported downward forcing the flow to descend and creating a mean downdraft near the inflow boundary as it can be observed in Fig 19a. Likewise, to maintain continuity, the flow needs to adjust the mesoscale profile at the outflow boundary (which resembles the inflow profile). Here, the horizontal flow needs to be decelerated which causes a mean updraft close to the outflow boundary. We note that these mean up- and downdrafts are most pronounced at nighttime, ranging between $0.1$–$0.4\,\mathrm{m\,s^{-1}}$ in this setup. Even though at daytime

10 these mean up- and downdrafts cannot be detected visually in the instantaneous vertical wind speed (see e.g. Fig. 12), we also found mean up- and downdrafts near the in- and outflow boundary but with a lower amplitude in the order of $0.05$–$0.1\,\mathrm{m\,s^{-1}}$.



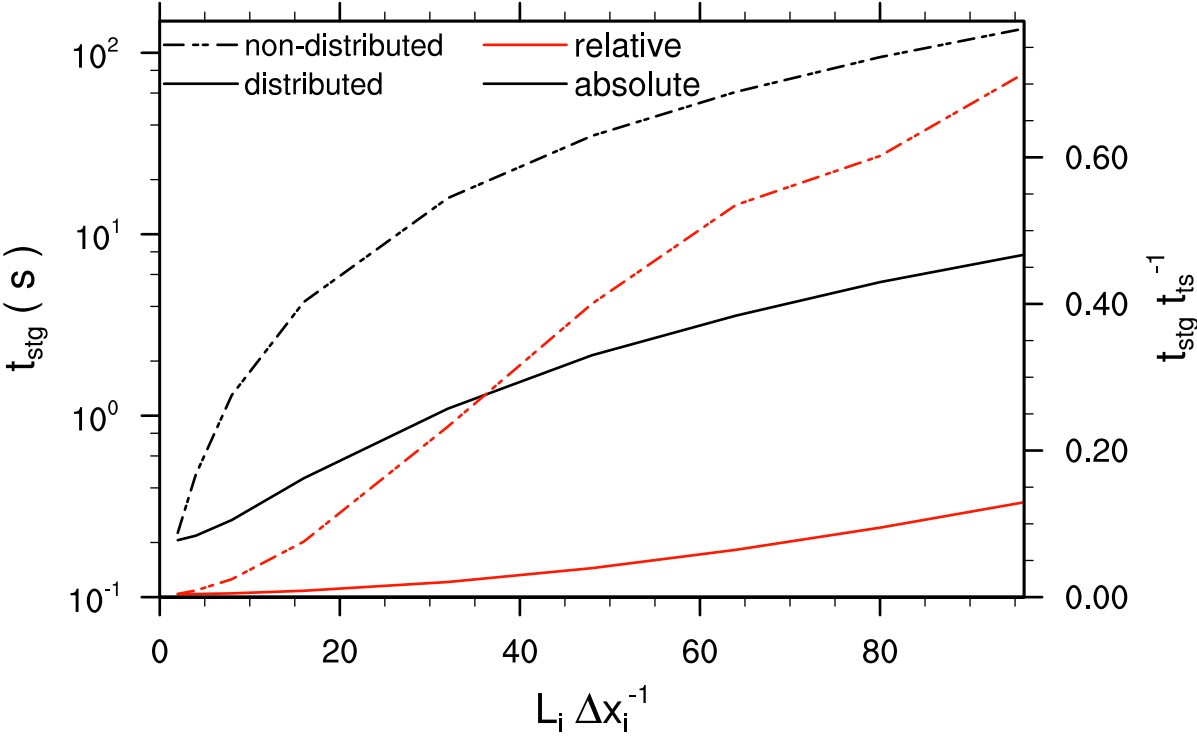

**Figure 20.** Consumed CPU-time by the synthetic turbulence generator for different turbulent length scales. The constantly prescribed length scale is normalized by the isotropic grid spacing. The left ordinate (black lines) shows the absolute consumed CPU time by the synthetic turbulence generator, while the right ordinate (red lines) shows the relative contribution with respect to the consumed CPU time spend for the time-stepping (i.e. without initialization, data output and finalization).

### 4.6 Computational efficiency of the synthetic turbulence generator

With respect to the mesoscale nesting, the generation of synthetic turbulence means the computationally most expensive part, while setting the boundary conditions and enforcing a divergence flow field is computationally much less expensive. In order to examine the efficiency of the turbulence generator implementation and estimate its computational cost, we have carried out

5    a performance test.

The most expensive part of the turbulence generation is the computation of the filtered random numbers, see Eqs. (27) and (28), which depend on the turbulent length scales. Therefore, based on the described setup in Sect. 3, we ran a set of idealized simulations where we varied the turbulent length scales. The length scales were set to a vertically constant value up to $z = 2500\,\mathrm{m}$, and to zero above. We performed 100 time integrations with a time step that was held constant at $0.5\,\mathrm{s}$

10   for all simulations. The simulations with different length scales were performed twice, one set where the computation of the filtered random numbers is distributed over all available MPI-processes (hereafter referred to as distributed), and a second set where the filtered random numbers are only computed on boundary MPI-processes (hereafter referred to as non-distributed)





and the rest of the MPI processes is on hold. The number of MPI-processes used for this scaling test was $n = 1296$. Figure 20 shows the consumed CPU time by the synthetic turbulence generator for different length scales. As expected, the consumed CPU time increases with increasing length scale. For small $L_i/\Delta x_i$ the consumed CPU time for both, non-distributed and distributed computation, is below 1% of the CPU time consumed in the time-stepping (see red lines) and no significant dif-

ference between both cases can be observed (black lines); this also shows that the additional MPI-communication required in the distributed case does not deteriorate the computational performance of the turbulence generator for small $L_i/\Delta x_i$. For larger $L_i/\Delta x_i$, the turbulence generator consumes significant portions of the available resources in the non-distributed case with relative contributions of >60% for length scales of about 2000 m ($L_i/\Delta x_i = 100$). In contrast, in the distributed case the CPU-consumption only increases moderately, reaching only up to 10% of the total CPU time consumed in the time-stepping

for length scales of about 2000 m. This shows that parallelizing the tasks needed for the synthetic turbulence generation saves significant computational resources.

Finally, we note that for the simulation covering an entire diurnal cycle, the length scales vary significantly with lower values at nighttime and larger values around noun. For these simulations the turbulence generation consumed about 2.5% of the CPU time.

**5  Summary and conclusions**

In this paper we presented a mesoscale-nesting interface for the PALM 6.0 model system that extends PALM's capabilities to simulate atmospheric boundary layers under evolving synoptic conditions. The mesoscale-nesting interface, which currently relies on output of the COSMO model, consists of two components: (i) the preprocessing interpolation tool INIFOR which provides initial and boundary conditions as a netCDF file, and (ii) PALM's internal boundary condition routines which

read and process the initial and boundary conditions as well as imposed additional synthetically turbulent fluctuations. We described INIFOR's interpolation methodology in detail, beginning with the relevant model differences between PALM and COSMO, leading to the conceptual steps needed to interpolate COSMO model output onto the PALM grid. Since the interpolated mesoscale boundary conditions are essentially free of turbulent fluctuations, the flow first needs to spatially develop before the turbulent transport of momentum, energy and water can be analysed. In order to minimize the extent of development

zones near the lateral inflow boundaries which the LES model would otherwise require to generate turbulence via shear and convective instabilities by itself (Mirocha et al., 2014), we employed a synthetic turbulence generation method according to Xie and Castro (2008). Using this approach, spatially and temporally correlated fluctuations of all three velocity components are generated based on parametrizations of the Reynolds stresses as well as turbulent length- and time-scales.

We demonstrated the nesting interface and the effectiveness of the synthetic turbulence generation using a semi-idealized

benchmark case: we simulated a convective boundary layer developing near Berlin, Germany, on a clear-sky spring day using initial and boundary conditions derived from DWD's operational COSMO-DE analysis. For the sake of analyzing the spatial development of the flow, the case was idealized in that we assume flat terrain with homogeneous grass land instead of using more realistic land-surface heterogeneity, in order to disentangle turbulence built-up due to the synthetic turbulence generation





and convective and shear instabilities from effects of the particular surface heterogeneities of the Berlin area. We found that the flow rapidly develops up- and downdrafts that are in equilibrium, whereas the adjustment of the TKE takes more distance of about $1.5\,u_\mathrm{h}z_\mathrm{i}/w_*$. The turbulent flow needs a fetch length that corresponds to more than one eddy turn-over time to be fully adjusted, which is similar to the observed adjustment distances reported by Muñoz-Esparza et al. (2015); Muñoz-Esparza

and Kosović (2018) with the cell perturbation method. Even though the adjustment distance could be significantly reduced, it is still in the order of several kilometers, which means that significant parts of the computational resources are still required only for the spatial development of the flow. To further reduce the computational effort, an alternative could be to combine the mesoscale nesting together with PALM's self-nesting (Maronga et al., 2020; Hellsten et al., 2020), i.e. a relatively coarse grid resolution in the outermost parent domain and finer grid resolutions within the nested child domains.

In our benchmark case, the boundary-layer in the mesoscale COSMO model does not develop synchronously with the boundary-layer in the LES. For example, in COSMO the boundary layer develops more rapidly before noon and the evening transition starts earlier compared to the LES simulation. As the signals due to non-synchronous boundary-layer development are imposed to the inflow boundary, these propagate through the LES domain creating a horizontally heterogeneous boundary layer during the transition phase. Furthermore, we observed under-resolved convective rolls emerging in the mesoscale model

that, similar to Mazzaro et al.'s (2017) findings, propagate into the LES domain. In the present study, we eliminated these roll-like structures by averaging over the inflow boundary, being aware that especially for larger domains synoptic-scale horizontal gradients or effects of mesoscale topography cannot be considered (Mazzaro et al., 2019). Alternatively, filtering the boundary conditions using a filter width corresponding to the horizontal grid spacing of the mesoscale model may help to eliminate such spurious flow structures.

Overall, especially the non-synchronous boundary-layer development and the imposed roll-like convection emphasize that the representation of the boundary layer in the LES and accompanied vertical gradients of wind velocity, potential temperature, etc. depend on the boundary-layer representation in the mesoscale model. If the boundary layer is not well captured in the mesoscale model, e.g. due to misrepresented convection and turbulent mixing, cloud cover, or atmosphere-surface exchange, the boundary layer resolved in the LES will also be affected by this. In such cases the physically more credible LES solution

(with respect to the boundary-layer representation) will be nudged towards the mesoscale solution. Here, further research is required to better understand the causes for such model discrepancies, under which circumstances they arise, and what the implications of the resulting nudging are for the turbulent boundary-layer flow.

Further branches of future development will be to enable INIFOR to also process WRF (Skamarock et al., 2008) and ICON (Zängl et al., 2015; Reinert et al., 2020) output, as well as to add further prognostic quantities to the mesoscale nesting

interface, e.g. chemical compounds, aerosols, liquid and frozen water. This is especially important to properly consider clouds and precipitation in the LES, which in turn also affect the surface net radiation and thus the entire boundary-layer development. However, we expect that in many future applications with mesoscale nesting the outermost parent domain will only run with relatively coarse grid resolution, so that cloud physics will not be captured well in the LES, especially for high-altitude clouds. Hence, we also plan to enable INIFOR to also provide incoming short- and longwave radiation fluxes, which, to date, PALM

is already enabled to consider either from observations or manually from observations or mesoscale model output.

The main focus of this study was to demonstrate the capability of the mesoscale nesting approach and to confirm the effectiveness of the synthetic turbulence generation to reduce the fetch length needed for the model to develop balanced turbulence characteristics. Dedicated evaluation runs of the PALM 6.0 model system including the mesoscale nesting interface are currently on its way within the project [UC]² (Scherer et al., 2019).

*Code and data availability.* The PALM model system 6.0 is freely-available at http://palm-model.org and distributed under the GNU General Public License v3 (http://www.gnu.org/copyleft/gpl.html). The preprocessor INIFOR is included in the PALM software repository as a utility and is currently available at https://palm.muk.uni-hannover.de/trac/browser/palm/trunk/UTIL/inifor. The simulations documented in the present article were performed using revision 4564 of the PALM model system, which includes INIFOR version 1.4.15. A complete archive of the software used for this publication, including the input data used, analysis and plotting scripts, as well as a step-by-step
reproduction guide is available at https://doi.org/10.25835/0084787 (Kadasch and Sühring, 2020).

*Author contributions.* EK: conceptualization and development of INIFOR, writing parts of the manuscript, conceptualization of the study, analysis of the results; MS: conceptualization and implementation of mesoscale nesting into PALM, adaption of the synthetic turbulence generator, writing parts of the manuscript, conceptualization of the study, analysis of the results; TG: adaptation of synthetic turbulence generator and writing parts of the manuscript; SR: conceptualization of INIFOR and mesoscale nesting; all authors: revision of the manuscript.

*Competing interests.* The authors declare that they have no conflict of interest.

*Acknowledgements.* This work is part of the [UC]² project. Financial support was provided by the German Federal Ministry of Education and Research (BMBF) under grant 01LP1601 within the framework of Research for Sustainable Development (FONA; www.fona.de), which is greatly acknowledged. All simulations with PALM have been performed on the supercomputers of the North German Supercomputing Alliance (HLRN). We thank Heike Schau-Noppel for her valuable comments on this manuscript.





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
