# Peer review of "Mesoscale nesting interface of the PALM model system 6.0"

_Geoscientific Model Development, 2020_

## Referee Comment (RC1) · Anonymous Referee #1 · 19 Jan 2021

This article describes the development of a preprocessor (INIFOR) that enables nesting of the PALM LES model within the mesoscale model COSMO. Besides the required coordinate transformations and interpolations required to provide boundary conditions for PALM, which are then read in as the netCDF files, the authors describe two additional steps performed within PALM that consist of an approach to remove residual divergence required for their incompressible LES solver, and a method to superimpose synthetic turbulence fluctuation into the smooth mesoscale velocity field from COSMO, employing the method from Xie & Castro (2008). The mesoscale-LES nesting capability is demonstrated with an idealized diurnal cycle, with a focus on the effectiveness of the synthetic turbulence generator. The article is well written and organized, and contains a good level of details required to understand the specific model development

and subsequent capability being developed. However, there are a number of important aspects that need to be addressed before the manuscript can be considered for publication in Geoscientific Model Development. My comments are listed here below in order of appearance in the manuscript:

1) Page 3, line 4. "Muñoz-Esparza et al. (2014) implemented the" should be "developed and implemented".

2) Page 3, lines 9-10. Where is the 15 km taken from? It seems to out of context. Please be more specific. Fetches of $\sim$5 km are typically reported for convective ABLs in Munoz-Esparza and Kosovic (2018).

3) Page 3, lines 15-17. This is an incorrect statement. Even in neutrally stratified conditions, there is thermal variance, although small. The fact that the cell perturbation method uses thermal effect to break up the two-dimensionality of the incoming flow does not mean is not applicable to shear-driven cases. Munoz-Esparza et al. (2015) explains these aspects in the context of neutrally stratified ABLs, so please correct this statement. In addition, for SBLs, there is buoyancy suppression, but this does not mean that the cell perturbation method "changes the physics of turbulence generation" as the authors mention either.

4) Page 3, lines 25-26. Here again the authors seem to be confused with what the "physics of turbulence" is and what the inflow turbulence techniques provide. The synthetic method, even if "inspired" by some scaling arguments, do generate "artificial" turbulence, which is not consistent with either forcing or the discretized governing equations. I see the authors are attempting to justify their choice by this is a biased statement. It is sufficient to mention it is a different approach, so I would suggest removing this sentence.

5) Page 3, lines 28-29. Also, in fairness, you should mention the computational cost, and how that compared to the cell perturbation method you have previously mentioned.

6) Page 4, line 24. "stochastic" -> "synthetic".

7) Page 7, line 27. Is COSMO a truly fully compressible formulation or is there acoustic filtering being applied?

8) Page 16, line 11 to end of the page. I do not think this discussion is required. This is a nested capability, and the great majority of NWP and LES model practitioners understand the concept of nesting. If the authors are willing to keep the information in the manuscript, I would suggest moving it to an appendix, since it is not part of the main body of content of the paper.

9) Page 18, Eq. 26. The use of this equation for convective conditions is highly questionable. The assumption that turbulent velocity correlations can be approximated by an exponential function is only reasonable for neutrally-buoyant shear flows. The presence of stability effects, breaks this assumption, so the use of a 2D synthetic flow field does not appear to be justified. The authors need at least to mention this strong limitation, and clearly acknowledging in the paper that they are using a method that is not designed for what they are using it for.

10) Page 20. These parameterizations of velocity variances from Brost ewt al. (1982) are for offshore boundary layers. How do you justify this choice? Such limitation needs to be acknowledged.

11) Page 23, Figure 7. I would suggest including wind direction profiles to help the reader appreciate the amount of directional variability (as described in the text).

12) Page 23, lines 5-6. Terrain is a key aspect in this type of nested simulation, where the terrain resolution changes drastically between the mesoscale and LES domains. Even if the authors use a flat terrain for their demonstration simulation, they should comment on how different terrain and atmospheric profiles are matched at the lateral boundaries of the LES domains.

13) Page 24, lines 14-16. This step does not seem to be necessary since you know

the mesoscale time-varying forcing.

14) Page 25, lines 10-11. Please define explicitly what the inner part is, both location and extent.

15) Page 25, lines 18-19. This could be related to the lack of turbulence in the residual layer.

16) Page 26, figure 9. RES does not seem to be defined. Do you mean REF?

17) Page 26, lines 3-6. It would really helpful to plot vertical profiles of theta and other quantities very near to the LES domain boundaries. This may shed some light into how PALM decreases temperature from the boundary forcing.

18) Page 26, line 15. It should be "W m-2".

19) Page 27, lines 5-6. Then, what is the issue? Looks like there is some imbalance, likely occurring near the lateral boundaries. This needs to be further investigated as it is a key aspect of the coupling between mesoscale and LES models.

20) Page 27, lines 9-10. This is confusing. Higher than what?

21) Page 27, lines 23-26. This can be prevented by truly embedding the LES domain within the mesoscale solution. I suggest the authors do so and report on their findings.

22) Page 27, lines 33-34. See previous comment. I believe this is due to your specific forcing settings.

23) Page 27, line 34 to end of the paragraph. It is unclear what the authors are trying to covey in this paragraph. Please rephrase.

24) Page 31, line 6. Why wouldn't you use u*, w* and H0 from the mesoscale too? These turbulent properties are not going to be reliable in the inflow region, as your resolved turbulence is not yet spun up.

25) Page 32, lines 3-4. There is a spurious kink toward the top of the ABL, likely

induced by the 'fading function' that does not look very smooth or reasonable. This aspect should be explicitly mentioned.

26) Page 32, line 16. This is not representative of what Fig. 14a shows, which is more ∼25 km to somewhat stabilize (and not even at all heights). I suggest the authors quantify the fetch. This can be done by using the last 10 km of the domain, where the solution looks stabilized, and use the average over that region as the 'target'. Then, you define equilibrium when you are stably within a 10% of that value.

27) Page 32, lines 18-19. This value is again highly biased and underestimated. At least 15 km are required. Please see my previous comment.

28) Page 35, line 5. If they are too energetic and keep varying their TKE with fetch, one cannot claim these structures have a reached a quasi-equilibrium state.

29) Page 34, lines 11-14. It is not until the surface reaches equilibrium that the flow can do that, since forcing at the surface is evolving, so it makes sense TKE is delayed compared to surface properties.

30) Page 34, line 23. Munoz-Esparza & Kosovic (2018) propose Uzi/w* as the parameter that indicates when inflow turbulence does not make any difference vs progressively increased fetches. It would be appropriate to mention that here.

31) Page 34, lines 25-26. This number is significantly underestimated. Please correct.

32) Section 4.4. Given the presence of under-resolved convective structures in the mesoscale solution, the 1-h time frequency of the lateral forcing mentioned earlier in the manuscript seems insufficient. The authors likely need to make that ∼1 min and rerun the simulation.

33) Page 35, last line. The estimated wavelength is 2-4delta, which seems too small to be resolved given the effective resolution of the fifth-order upwind advection scheme used by the authors. Could the authors describe how is the wavelength estimated?

34) Page 39, lines 21-23. Munoz-Esparza et al. JAMES2017 discusses a way to eliminate these structures, and that would be pertinent to mention here.

35) Page 40, Figure 19. Please include wind speed and potential temperature contours as well. This may help you diagnosing the issue with the LES over-cooling.

36) Page 40, lines 6-7. Is this a result from the divergence-free adjustment not being totally effective? Please comment on this.

37) Page 43, lines 1-3. Again, this should be updated to report a more realistic value according to the presented results. It is more $\sim$3.0.

38) Page 43, lines 4-5. Munoz-Esparza et al. (2015) showed on a apples to apples comparison (i.e., same LES model, forcing, etc) that the cell perturbation method required shorter fetches compared to Xie & Castro (2008). This should be mentioned. Also, for convective conditions, cell perturbation results reported in Munoz-Esparza & Kosovic (2018) are smaller than 2.0uhzi/w*, which is shorter than required fetches presented herein, more $\sim$3uhzi/w*. I agree these can be called 'similar', but there are considerable differences that deserve to be mentioned. Also, the behavior of the fetch development with the cell perturbation is more systematic and have been show to produce well equilibrated solutions, while here there is for certain cases a lack of development. The authors need to mention this aspect.

39) Page 43, line 20. This is likely caused by the lateral boundary conditions that do not match mesoscale variability (i.e., they are uniform in space and do not change between the LES domain boundaries).
* * *

---

## Short Comment (SC1) · 5 Mar 2021

The parent domain (COSMO) has a resolution of 2.8 km and 2.2 km, while the LES nested domain(PALM) is of the resolution 25 m. In the case of WRF, the ratio is optimised to be 3:1. Is there any such condition for PALM:COSMO ?

---

## Referee Comment (RC2) · Anonymous Referee #2 · 9 Mar 2021

The manuscript "Mesoscale nesting interface of the PALM model system 6.0" by Kadasch et al. presents newly developed mesoscale nesting capabilities of the large-eddy simulation model PALM within the COSMO mesoscale model. The preprocessor INIFOR is developed to provide proper initial and lateral boundaries conditions, accounting for the differences between model variables and coordinate systems. A synthetic turbulence generator is also implemented to accelerate the transition between mesoscale and turbulence-resolving LES-scale. A semi-idealized test case of a diurnally varying dry boundary layer is presented, mostly to demonstrate the effectiveness of the synthetic turbulence generator. I applaud the authors' efforts as this could greatly expand the usefulness of the PALM model especially for real cases. The manuscript is well written, with details of the model clearly presented, and rationales thoroughly

explained. I am not sure if GMD has a word limit, but given the nature of a journal article, the authors might consider consolidating the manuscript. Overall, I suggest minor revisions.

Major comments:

1. In a nested model setup, a sponge zone or a damping layer is commonly adopted at the lateral boundaries of the nest, such that spurious wave reflections due to changes in grid resolution across the nest interface are absorbed. Correct me if I am wrong, but in the proposed implementation, Dirichlet boundary conditions is adopted to drive the PALM model. Would this cause numerical issues at the lateral boundaries, especially the outflow boundary? It would be hard to observe spurious reflections at the lateral boundaries under convective conditions, it would be much easier to spot numerical waves under nighttime stable conditions. Also, could the absence of a sponge layer at least partially explain the "rim" of time-averaged vertical motions along the lateral boundaries presented in Fig. 19? If implementation of a sponge zone is out of the scope for the current model, the authors should at least provide some justification for using Dirichlet lateral boundary conditions.

2. Does COSMO have LES capability? If so, is the LES closure in the COSMO model the same as that in the PALM model? If the answer is also yes, I would suggest the authors try the following experiment, to help diagnose some of the issues such as the mismatch of potential temperature profiles and wind profiles. The authors could set up a similar LES domain within the COSMO model, and run the same test case within COSMO. (I assume COSMO has one-way nesting capability). The differences due to the model coordinate systems should not matter too much given the limited horizontal extent of the LES domain. Then the authors could compare the nested COSMO results with the nested PALM results to understand, for example, the influence of different land-surface schemes on the vertical wind and temperature profiles in the nested domain.

Minor comments:

1. Page 3, Lines 25-26, I have to disagree with this statement. Both the synthetic turbulence method and the cell perturbation method add "artificial" perturbations to the flow that are not strictly consistent with the "physics of turbulence production". Also, I believe the cell-perturbation method is also capable of allowing turbulence to "freely develop depending on the mean-gradients of potential temperature and wind speed". I would like to hear your explanation but I don't think one method has an advantage over the other in terms of physics.

2. Page 5, please combine the first paragraph with the last single-sentence paragraph on Page 4.

3. Page 5, line 5, "Both nesting features may, however, may be . . .", please fix the grammar.

4. Page 7, line 10, better "do not generally" than "do generally not".

5. Page 7, lines 9-17, so the divergence is removed at the LES domain level, rather than at each grid point, is that right? Perhaps point this out explicitly.

6. Page 18, perhaps the authors will explain later, but how is Eq. 17 implemented near boundaries, where points outside the computational domain are required in the double summation?

7. Page 19, lines 14-15, is there a reason why "at opposite boundaries (west and east, as well as north and south) we use the same $\Psi i$" ?

8. Page 19, lines 22-23, correct me if I am wrong, but 2d domain decomposition means that the domain is split in the x and y directions, right? So why would this enable Eq. 27 to be computed locally, so that "no global communication is necessary". For example, on the west boundary, the summation still requires information across processors in the y-direction, right?

9. Page20, lines 24-25, so these parameterized Reynolds stresses apply only to un- stable conditions? What about stable and/or neutral conditions? Did you drop the MO

term?

10. Page 21, correct me if I am wrong, but zi is obtained by horizontal averaging along the "boundary grid point", but u* is obtained by horizontal averaging "within the model domain", why the difference?

11. Page 22, Eq. 37, the turbulent time scale appears to be a dimensionless number, how is it transformed into actual time?

12. Page 24, Line 12, "indicate" rather than "indicates".

13. Page 24, Line 25, "subsequently" rather than "in the subsequent",

14. Page 26, Line 7, you meant "RES"?

15. Page 26, Line 18, how do you set the prescribed values of H0 and LE0 ?

16. Page 32, Line 13, I would intuitively expect a monotonic increase of resolved TKE from the coarse grid to the fine grid, approaching some asymptotic values inside the LES domain. But why the TKE peak?

17. Page 40, Lines 5-8, this are most likely compensating vertical motions due to horizontally divergent and convergent flow at the inflow and outflow boundaries, as a result of continuity. I would not over-interpret this like "horizontal momentum needs to be transported downward forcing the flow to descend".

18. Page 41, Fig. 20, please double check the legends, both "distributed" and "absolute" are marked with solid black lines.

19. Page 41, line 2, "is" rather than "means".
* * *

---

## Author Comment (AC1) · 4 May 2021

**Mesoscale nesting interface of the PALM model system 6.0 — Final author response**

We thank both reviewers for the thorough and thoughtful comments which drew our attention to weaknesses in the manuscript and helped us to improve it. Since two themes of criticism come up repeatedly, we want to address them here collectively before entering the more detailed discussion comment by comment.

The first one is the fact that we use hourly COSMO output for nesting PALM, which may contribute to a time-lagged representation of fast transitions and contribute to the generation of coherent turbulent structures due to a lack of change in the derived boundary conditions. We agree that, generally, it is desirable and that it would be beneficial to use higher-frequent COSMO output to derive PALM boundary conditions, ideally using every COSMO time step. However, the goal of the presented mesoscale interface, from the beginning, was to provide a tool for both scientists and practitioners (in municipalities and consultancies). The latter especially do not typically have the resources to carry out mesoscale simulations on their own and rely on external meteorological data sources. Thus, we decided to rely on operationally archived analysis and forecast model data by the DWD, which are only available on an hourly basis. Generally, our nesting approach is of course not limited to hourly forcing data, but for the presented benchmark simulation we decided to use such a setup.

The second theme of criticism concerns the lack of explanation for the differences between the boundary layer representation of COSMO and PALM, in particular the asynchronous morning and afternoon transitions. While we agree, these differences need to be analysed and understood, the scope of this manuscript is the technical description of our nesting methodology and proof of concept. As we are nesting two very different models with different modelling approaches for the boundary layer dynamics, we also expect discrepancies to some degree. We highlighted some of the differences between the models in order to raise the awareness of users to issues they might face and give suggestions on how to resolve them. However, we believe that an investigation into the reasons for the model differences is best done not with the presented simulation setup but with more idealized simulations where more parameters can be controlled. To emphasize this, we have mentioned this general motivation in the introduction.

In the following, we address each reviewer comment one by one with the reviewer comments in *italics* and our responses in regular type.

**Response to Reviewer 1**

*This article describes the development of a preprocessor (INIFOR) that enables nesting of the PALM LES model within the mesoscale model COSMO. Besides the required coordinate transformations and interpolations required to provide boundary conditions for PALM, which are then read in as the netCDF files, the authors describe two additional steps performed within PALM that consist of an approach to remove residual divergence required for their incompressible LES solver, and a method to superimpose synthetic turbulence fluctuation into the smooth mesoscale velocity field from COSMO, employing the method from Xie & Castro (2008). The mesoscale-LES nesting capability is demonstrated with an idealized diurnal cycle, with a focus on the effectiveness of the synthetic turbulence generator. The article is well written and organized, and contains a good level of details required to understand the specific model development and subsequent capability being developed. However, there are a number of important aspects that need to be addressed before the manuscript can be considered for publication in Geoscientific Model Development. My comments are listed here below in order of appearance in the manuscript:*

*1) Page 3, line 4. "Muñoz-Esparza et al. (2014) implemented the" should be "developed and implemented".*

We agree and changed the wording accordingly.

*2) Page 3, lines 9-10. Where is the 15 km taken from? It seems to out of context. Please be more specific. Fetches of ~5 km are typically reported for convective ABLs in Munoz-Esparza and Kosovic (2018).*

We have taken the 15-km value from Fig. 10 and Fig. 13 from Muñoz-Esparza and Kosović (2018), where cases U15H2 and U15H4 show even larger adjustment fetches in the vertical velocity variance, skewness, heat- and momentum flux. Nevertheless, we agree with the reviewer that adjustment fetches depend on the specific atmospheric setup and that naming specific values without providing more information is inappropriate. We have removed this statement from the text.

*3) Page 3, lines 15-17. This is an incorrect statement. Even in neutrally stratified conditions, there is thermal variance, although small. The fact that the cell perturbation method uses thermal effect to break up the two-dimensionality of the incoming flow does not mean is not applicable to shear-driven cases. Munoz-Esparza et al. (2015) explains these aspects in the context of neutrally stratified ABLs, so please correct this statement. In addition, for SBLs, there is buoyancy suppression, but this does not mean that the cell perturbation method "changes the physics of turbulence generation" as the authors mention either.*

We agree that this statement is a bit rough and needs to be more differentiated. Adding temperature fluctuations first produces turbulence in the vertical velocity component which then is transferred towards the horizontal components via the pressure-correlation term. In mostly shear-driven boundary layers this mechanism of turbulence production does not meet the

physics, where turbulence is produced (by shear) first in the horizontal components, which is then transferred towards the vertical component. However, we do not want to give the impression that the cell-perturbation method is not applicable to such situations. It has been proved in several studies that this method works well also in shear-driven boundary layers. This is because the buoyancy-produced turbulence gets rapidly transferred from the vertical to the horizontal components while temperature fluctuations also rapidly dissipate within the adjustment fetch.

To be fair, even the method presented by Xie and Castro (2008) does not fully meet the physics of turbulence production. For example, in buoyancy-driven boundary layers the driving temperature fluctuations that trigger the disturbances in the horizontal and vertical components, are missing. The added disturbances in the velocity components thus trigger temperature fluctuations, which is in fact also unphysical. To overcome this, also temperature perturbations need to be considered that correlate spatially with the perturbations imposed onto the w-component.

We have removed this statement from the manuscript.

*4) Page 3, lines 25-26. Here again the authors seem to be confused with what the "physics of turbulence" is and what the inflow turbulence techniques provide. The synthetic method, even if "inspired" by some scaling arguments, do generate "artificial" turbulence, which is not consistent with either forcing or the discretized governing equations. I see the authors are attempting to justify their choice by this is a biased statement. It is sufficient to mention it is a different approach, so I would suggest removing this sentence.*

As with the reviewer comments 2 and 3, we agree with this one and have removed the statement concerning unphysical turbulence production.

*5) Page 3, lines 28-29. Also, in fairness, you should mention the computational cost, and how that compared to the cell perturbation method you have previously mentioned.*

Indeed, the method by Xie and Castro (2008) is computationally more expensive compared to the cell-perturbation method, and — if not parallelized — the method is slower than the cell perturbation method. However, our implementation is highly parallelized across all MPI ranks so that the additional cost of the turbulence generation to the total computational expense is negligible as we show in Fig. 20. Thus, the computational demand was only a secondary criterion in this case. We agree that other implementations struggle with the computational cost of Xie and Castro's method, especially for large flow problems, e.g. Zhong et al. (2021). However, since this is not an issue with our implementation, we would like to omit this discussion in the introduction. Instead we added the following note about this in Section 2.3:

"Especially for non-parallelized implementations of the method (e.g. Zhong et al., 2021) this may become a limiting factor with respect to the computational demand."

*6) Page 4, line 24. "stochastic" → "synthetic".*

Done.

*7) Page 7, line 27. Is COSMO a truly fully compressible formulation or is there acoustic filtering being applied?*

COSMO solves a compressible system of equations, which includes sound waves. The problem of small time steps required to satisfy numerical stability in the presence of fast travelling phenomena such as sound waves are mitigated in COSMO using a time-splitting method where the fast terms related to sound propagation are integrated with a smaller time step than the rest of the system.

*8) Page 16, line 11 to end of the page. I do not think this discussion is required. This is a nested capability, and the great majority of NWP and LES model practitioners understand the concept of nesting. If the authors are willing to keep the information in the manuscript, I would suggest moving it to an appendix, since it is not part of the main body of content of the paper.*

We agree with the referee that the discussion would normally not be required. However, in the PALM overview paper by Maronga et al. (2020), we suggested the necessity of the geostrophic wind forcing with this nesting approach and we want to keep this information in the manuscript to clear up the misconception. We moved the section to the appendix, where — we agree — it fits much better.

*9) Page 18, Eq. 26. The use of this equation for convective conditions is highly questionable. The assumption that turbulent velocity correlations can be approximated by an exponential function is only reasonable for neutrally-buoyant shear flows. The presence of stability effects, breaks this assumption, so the use of a 2D synthetic flow field does not appear to be justified. The authors need at least to mention this strong limitation, and clearly acknowledging in the paper that they are using a method that is not designed for what they are using it for.*

Although an exponentially decaying autocorrelation is widely used in literature, e.g. to estimate random errors in observations (Lenschow et al. 1994), or in Lagrangian dispersion modelling (Weil et al., 2004), the reviewer is right with this point. An exponentially decaying autocorrelation function has been proved to be valid for shear-driven flows. To our knowledge, this has not been done for stratified flows and no universally valid formulation for all flow regimes exists in the literature. We added a remark on this in the revised manuscript below Eq. 26.

*10) Page 20. These parameterizations of velocity variances from Brost et al. (1982) are for offshore boundary layers. How do you justify this choice? Such limitation needs to be acknowledged.*

In the revised manuscript, we have made a remark on this. We have chosen the Brost parametrization due to lack of appropriate alternatives. Please correct us if we missed something, but to our knowledge no study exists which provides all reynolds-stress components and integral scales throughout the entire boundary layer inferred from one dataset. Brost et al. (1982) have shown that the turbulence budget within the boundary layer is not overly affected by the stratocumulus, meaning that the boundary layer is mostly surface driven. Further, the Brost parametrization has been used in Lagrangian dispersion modelling which in turn has been

successfully validated against observations under various stability regimes over land (Rotach et al. 1996).

For these reasons, we think that the Brost parametrization is an appropriate choice and can be universally applied to boundary layers. Nevertheless, we agree with the reviewer and discuss this in the revised manuscript.

*11) Page 23, Figure 7. I would suggest including wind direction profiles to help the reader appreciate the amount of directional variability (as described in the text).*

That is a good point, we added the wind direction in the plot.

*12) Page 23, lines 5-6. Terrain is a key aspect in this type of nested simulation, where the terrain resolution changes drastically between the mesoscale and LES domains. Even if the authors use a flat terrain for their demonstration simulation, they should comment on how different terrain and atmospheric profiles are matched at the lateral boundaries of the LES domains.*

In cases where the microscale column extends below the lowest COSMO cell and, thus, below COSMO's terrain, data is currently extrapolated as a constant. Beyond that, there is currently no further terrain adaptation made to blend the terrain from the coarse mesoscale into the fine microscale resolution. We agree with the referee that this information is missing from our manuscript and added a statement about this in section 2.2.4 at the end of paragraph "Three-dimensional interpolation".

We are currently exploring methods to transition the mesoscale terrain and/or atmosphere to the microscale terrain more elegantly. As the terrain mapping may have various implications for the flow solution and turbulence generation at the boundaries, we decided to defer a discussion about this to a follow-up paper and; in this manuscript, we only present our current approach of mesoscale nesting.

*13) Page 24, lines 14-16. This step does not seem to be necessary since you know the mesoscale time-varying forcing.*

Yes, in case of one-dimensional mesoscale forcing (one vertical profile is prescribed for all lateral boundaries) this is actually not necessary. For a heterogeneous forcing, however, the wind direction along the lateral boundaries can change (though only slightly for the considered model domain sizes of a few tens of kilometers). Hence, in order to account for this, we had done this backward-trajectory analysis, even though in the two-dimensional forcing the differences compared to just taking the mean-inflow profiles is very small in the considered case. We would like to keep this in for future reference, but made a note in the text that this is only relevant if strong horizontal gradients in the mesoscale forcing accompanied with significant changes in inflow wind direction occur.

*14) Page 25, lines 10-11. Please define explicitly what the inner part is, both location and extent.*

We have now indicated the averaging domain used to evaluate the inner-domain profiles in Fig. 12 and refer to this when we first mention the inner-domain profile. Moreover, we have indicated the location of the xz-cross-section (Fig.11) in Fig. 12.

*15) Page 25, lines 18-19. This could be related to the lack of turbulence in the residual layer.*

The residual layer, which is still present at 09:00 UTC in the morning, usually does not show much turbulence, or only small-scale turbulence produced by local shear as observations and numerical simulations show (e.g. Blay-Carreras et al. 2014, Fig. 7a and associated text). Usually the stably-stratified layer vanishes when the growing boundary layer reaches the residual layer, while the boundary layer heats-up by surface heating and entrainment processes. (The latter are due to wind shear and the penetrating thermals triggered by surface heating.) To our knowledge, turbulence within the residual layer does not contribute significantly to the erosion of the stably-stratified layer from beyond.

However, one has to keep in mind that the residual layer and its characteristics are also advected from upstream locations. Another reason could be that further upstream, the boundary-layer development in the morning hours has been further advanced and that this already developed boundary layer has been simply advected into the PALM domain. We added this hypothesis to the discussion.

As we wrote in our introductory remark to this response, we unfortunately do not have highly resolved COSMO data available to further investigate this issue which we think would extend beyond the scope of this manuscript for the reasons mentioned.

*16) Page 26, figure 9. RES does not seem to be defined. Do you mean REF?*

Yes, indeed, we corrected the label.

*17) Page 26, lines 3-6. It would really helpful to plot vertical profiles of theta and other quantities very near to the LES domain boundaries. This may shed some light into how PALM decreases temperature from the boundary forcing.*

Figure 11 provides this information already. Figure 11a shows a xz-cross-section of the potential temperature and corresponding boundary layer depth, with the right-hand side being the inflow boundary. The boundary-layer depths as well as the temperature decrease gradually with increasing distance to the inflow boundary. This indicates that warmer air is advected into the LES domain rather than there being a sharp drop in potential temperature close to the boundary. The fact we observed the same behavior (COSMO significantly warmer than PALM at 10:00UTC) in the test simulation PSF, where we prescribed the surface forcing identical to COSMO, also supports this.

Unfortunately we cannot dig deeper into this issue and into what happens in COSMO between 9-10UTC within the scope of the paper.

*18) Page 26, line 15. It should be "W m-2".*

Fixed. Thanks for pointing this out.

*19) Page 27, lines 5-6. Then, what is the issue? Looks like there is some imbalance, likely occurring near the lateral boundaries. This needs to be further investigated as it is a key aspect of the coupling between mesoscale and LES models.*

To explain the temperature increase in COSMO, we have made a bulk estimation of boundary-layer evolution as presented in Stull (1988, page 457). According to this, a sensible heating of about 350 W/m2 would have been required to explain the boundary-layer development during 9-10 UTC in COSMO, while the actual sensible surface heating at that time was only 100-120 W/m2. Even increased entrainment due to residual-layer turbulence would not explain the missing energy. Another reason could be local advection of a warmer and deeper boundary layer from upstream locations, though the boundary layer at 9 and 10 UTC upstream looks pretty similar.
Please also see our introductory remark on model differences at the beginning of our reply.

*20) Page 27, lines 9-10. This is confusing. Higher than what?*

We agree, this was not well explained and rephrased the sentence.

*21) Page 27, lines 23-26. This can be prevented by truly embedding the LES domain within the mesoscale solution. I suggest the authors do so and report on their findings.*

We assume by "truly embedding" the reviewer either refers to (i) nudging the LES solution towards the mesoscale one in addition to setting time-dependent Dirichlet boundary conditions or to (ii) coupling the boundary conditions at a higher frequency to COSMO. For the latter, please see our introductory remark at the beginning of our reply.

While we agree a nudging approach could reduce the horizontal heterogeneity in the model domain, this would (i) very likely not improve the quality of the simulation in the internal domain and (ii) have the same time-lag problems with hourly forcing data as using boundary conditions.

As for the first point, given the differences in filtering of the equations, parameterization of turbulence, and numerical resolution, PALM's representation of the (dry convective) boundary layer is generally more trustworthy than COSMO's. Thus, we believe, nudging PALM's solution towards COSMO's would rather remove the gain in quality than improve it. If COSMO would be able to represent the boundary layer as well as PALM, there would be no need to run an LES model in the first place. Because the two models follow such different modelling approaches, we should expect discrepancies between their solutions. But to maintain the strengths of the LES, we should allow it to develop a balanced solution using buffer regions and not draw it towards the less accurate mesoscale solution.

As for the second point, both boundary conditions and nudging profiles based on linearly interpolated hourly forcing data will suffer from a time lag and temporal smoothing of fast transitions in the host model. For instance, if the morning transition on a given day occurs

between 15 and 45 minutes past the hour, the morning transition as represented by hourly COSMO analyses will be a smooth one-hour transition instead of a sharper 30-minute one.

As for our reasons to use only hourly forcing data, please see our introductory remark to our answers.

*22) Page 27, lines 33-34. See previous comment. I believe this is due to your specific forcing settings.*

In the manuscript, we have excluded the LES surface forcing to be the reason for the not fully synchronous boundary-layer evolution by running the PSF simulation where the fluxes (taken from COSMO) were explicitly prescribed in the LES. Further, as we have explained in our reply to the referee's comment 19, COSMO's surface forcing also cannot explain the boundary-layer development between the morging transition – significantly larger sensible surface heating would have been required for this.

*23) Page 27, line 34 to end of the paragraph. It is unclear what the authors are trying to covey in this paragraph. Please rephrase.*

The reviewer is right, the paragraph was not well phrased. We wanted to emphasize that the LES solution is continuously pushed towards the mesoscale solution, independent of whether the mesoscale solution is biased or not. As we cannot give an explanation as to what caused the rapid development of the COSMO boundary layer during the morning transition and we do not want to make preliminary conclusions that this is a model bias in the mesoscale model, we decided to keep this paragraph out.

*24) Page 31, line 6. Why wouldn't you use u, w and H0 from the mesoscale too? These turbulent properties are not going to be reliable in the inflow region, as your resolved turbulence is not yet spun up.*

In fact, taking surface quantities directly from the mesoscale model was our original idea. However, the friction velocity, Obukhov length, etc., depend on the surface properties which are aggregated over mesoscale grid cells which often only include a rough representation of the different surface types. Further, buildings and induced drag are often not considered as urban parametrizations are often not enabled in operational datasets. As the degree of detail is higher in the fine-scale LES setup (terrain, buildings, surface types, plants) and we want to generate synthetic turbulence roughly reflecting the mean conditions within the LES domain, we think taking surface properties from the LES domain is more accurate and universal than taking them from the mesoscale model.

*25) Page 32, lines 3-4. There is a spurious kink toward the top of the ABL, likely induced by the 'fading function' that does not look very smooth or reasonable. This aspect should be explicitly mentioned.*

In the revised version, we now explicitly mention the kink at the boundary-layer top and that this is caused by the fading function. Usually turbulence vanishes above the boundary-layer top, so

that the vertical gradient in the Reynolds stress components would be even stronger without the fading function. Although the profile does not look smooth beyond the boundary-layer top, we think this does not cause any harm since the length scales are also faded. With this, the imposed turbulence becomes more and more small-scale and will quickly dissipate within the stably-stratified free atmosphere. We added this also to the revised manuscript.

*26) Page 32, line 16. This is not representative of what Fig. 14a shows, which is more ˜25 km to somewhat stabilize (and not even at all heights). I suggest the authors quantify the fetch. This can be done by using the last 10 km of the domain, where the solution looks stabilized, and use the average over that region as the 'target'. Then, you define equilibrium when you are stably within a 10% of that value.*

In the original manuscript, we estimated this visually, probably with a tendency to underestimate the required fetch. We have quantified the fetch as suggested by the reviewer, which we now describe in the text, and adapted the values accordingly. We now also mention the height dependency of the adjustment explicitly.

*27) Page 32, lines 18-19. This value is again highly biased and underestimated. At least 15 km are required. Please see my previous comment.*

Following the reviewer comment 26, we have adapted the numbers; we also revised the given dimensionless value in the following sentence.

*28) Page 35, line 5. If they are too energetic and keep varying their TKE with fetch, one cannot claim these structures have a reached a quasi-equilibrium state.*

Yes, that's true. By "equilibrium" we wanted to express that coherent structures develop which do not change their area-size distribution any more, i.e. that the fraction of up- and downdrafts becomes constant. But we agree that "equilibrium" is the wrong term here and rephrased the sentence. We used a similar phrasing in the second paragraph of our "Summary and conclusions", which we also changed.

*29) Page 34, lines 11-14. It is not until the surface reaches equilibrium that the flow can do that, since forcing at the surface is evolving, so it makes sense TKE is delayed compared to surface properties.*

In the revised manuscript, we mention this aspect, i.e. that surface adjustment is a necessary prerequisite for the turbulent flow adjustment.

*30) Page 34, line 23. Munoz-Esparza & Kosovic (2018) propose Uzi/w\* as the parameter that indicates when inflow turbulence does not make any difference vs progressively increased fetches. It would be appropriate to mention that here.*

In the revised manuscript we now mention Munoz-Esparza & Kosovic (2018) who used a similar form of convective scaling to correct the amplitude of the imposed temperature perturbations. We

are sorry that we have missed this, it was not our intention to disregard their findings and argumentations.

*31) Page 34, lines 25-26. This number is significantly underestimated. Please correct.*

We have revised the values throughout the entire manuscript accordingly. Furthermore, we omit the direct comparison against the cell perturbation method in the next sentence, as we think that it does not contribute much to the manuscript when we compare results from our method against the cell perturbation method based on different cases.

*32) Section 4.4. Given the presence of under-resolved convective structures in the mesoscale solution, the 1-h time frequency of the lateral forcing mentioned earlier in the manuscript seems insufficient. The authors likely need to make that ~1 min and rerun the simulation.*

We agree with the reviewer that one-hourly coupling is probably not sufficient and higher coupling frequencies are desirable. Ideally, spatial resolution and forcing frequency should match each other, i.e. spatial resolution should be less than or equal to u*dt_forcing. Independent of the forcing frequency, spurious under-resolved structures, numerical oscillations, etc., that occur in the mesoscale solution may enter the LES domain. However, in the mesoscale model, these structures may drift or decay faster than can be represented with a one-hourly coupling. With a one-hourly coupling such structures may persist over a longer time interval and may bias the flow accordingly, which would not occur for higher coupling frequencies. In the revised manuscript, we reworked the end of section 4.4 and added a brief discussion on effects of the coupling frequency.

To circumvent this problem, one should either increase the forcing frequency or spatially filter the boundary conditions — thus reducing the effective resolution — such that the coupling time step corresponds to the spatial scales. With our homogeneous case, we do an extreme but simple version of the latter by horizontally averaging the boundary conditions. In both our original and the revised manuscript, we thus suggest to the reader to prefer using these homogeneous boundary conditions over the directly interpolated ones to mitigate this problem.

Repeating the simulations with higher coupling frequencies, however, is not possible for us as COSMO analysis data is only available in hourly intervals and we do not have the resources to carry out a separate COSMO run to provide high-frequency output data on our own. But we added a discussion to the text.

*33) Page 35, last line. The estimated wavelength is 2-4delta, which seems too small to be resolved given the effective resolution of the fifth-order upwind advection scheme used by the authors. Could the authors describe how is the wavelength estimated?*

We have estimated the wavelength just visually, mainly by means of the structures on the left-hand side of Fig. 18c, which we now also mention in the manuscript.
The reviewer is right with his comment. COSMO also employs the 5th-order Wicker and Skamarock scheme along the horizontal dimensions, meaning that small scale structures would rapidly dissipate. According to this, any grey-zone turbulence would vanish quickly, or at least would be smoothed significantly.

On closer inspection, we think the estimated wavelength values are closer to 0.1-0.15 degree rather than 0.05 to 0.1 degree, so that we have revised the values also. These structures are then resolved by 4 to 6 grid points, which is still on the scales where the 5-th order scheme smoothes structures, though these do not instantaneously vanish.

As far as we know, no systematic analysis of grid-dependent grey-zone structures has been done so far for the COSMO model, so that we can only speculate that the occurrence of these small scale elongated structures is caused by similar reasons as reported by Ching et al. (2014) and Zhou et al. (2014) as shown for WRF. In the revised text we made this clear. Nevertheless, independent of the reason for these structures, they are heavily affected by the numerics (e.g. advection scheme) and are thus not reliable, thus, imposing them at the lateral LES domain boundaries with the current coupling timestep given causes several implications we want to emphasize with this paragraph.

*34) Page 39, lines 21-23. Munoz-Esparza et al. JAMES2017 discusses a way to eliminate these structures, and that would be pertinent to mention here.*

We thank the reviewer for pointing this out and added a comment on and reference to Munoz and Esparza et al.'s approach (2017) in our *Summary and conclusions* section. Since we do not control the entire model chain and can not modify COSMO in a similar way, we feel the remark fits better there as an aside than in the previous section.

*35) Page 40, Figure 19. Please include wind speed and potential temperature contours as well. This may help you diagnosing the issue with the LES over-cooling.*

Thank you for this suggestion. We agree, this would help shed more light into the issue, but we would like to omit such analyses in this manuscript for the reasons we laid out in our introductory remark.

*36) Page 40, lines 6-7. Is this a result from the divergence-free adjustment not being totally effective? Please comment on this.*

The divergence correction presented in Eq. 1 and 2 removes any remaining divergence in the boundary conditions. The correction is per definition 100% effective, i.e. the inflow into the model domain equals the outflow.. It, thus, ensures global mass-flux conservation.

The updrafts and downdrafts seen in Fig. 19 are local phenomena and are not affected by the global divergence correction. They also do not indicate any remaining local divergence. (Keep in mind that this is a cross section of a 3D flow.) Any local divergence, which almost certainly occurs in the predictor step of incompressible solvers, is removed in the pressure correction step maintaining the incompressible continuity equation.

The downdrafts near the inflow boundary actually result from maintaining local continuity where differences in the horizontal velocities between COSMO and PALM near the surface are balanced by vertical motions in PALM. The differences are attributed to greater surface friction in

COSMO compared to PALM. In other words, too much momentum enters the PALM domain at lower heights. The large discrepancy in surface friction at the inflow boundary is probably also connected to the not fully adjusted flow in PALM where the surface momentum transport near the inflow boundary undergoes an adjustment. At the outflow boundary, however, we can observe an updraft and a deceleration of the flow, again indicating more surface friction in the mesoscale model compared to PALM. We added a note on this at the end of the paragraph.

*37) Page 43, lines 1-3. Again, this should be updated to report a more realistic value according to the presented results. It is more ~3.0.*

We have adjusted the numbers accordingly. Furthermore, we removed the statement that this is "similar to the cell perturbation method" in the referred sentence.

*38) Page 43, lines 4-5. Munoz-Esparza et al. (2015) showed on an apples to apples comparison (i.e., same LES model, forcing, etc) that the cell perturbation method required shorter fetches compared to Xie & Castro (2008). This should be mentioned. Also, for convective conditions, cell perturbation results reported in Munoz-Esparza & Kosovic (2018) are smaller than 2.0uhzi/w, which is shorter than required fetches presented herein, more ~3uhzi/w. I agree these can be called 'similar', but there are considerable differences that deserve to be mentioned. Also, the behavior of the fetch development with the cell perturbation is more systematic and have been show to produce well equilibrated solutions, while here there is for certain cases a lack of development. The authors need to mention this aspect.*

We have mentioned this comparison in the revised text. However, we have removed the specific statement the reviewer refers to. We think that an evaluation at this point of which method is better in terms of adjustment fetches under evolving conditions is not target-aimed and even it is not fair. This is because Munoz-Esparza & Kosovic (2018) investigated idealized situations with quasi stationary boundary layers where the boundary conditions themself do not undergo any transition, whereas in our case only an evolving boundary layer was investigated. Evolving atmospheric conditions, however, complicate the analysis and lead to a horizontally heterogeneous boundary layer which also affects the turbulence adjustment fetch, which then is always time-lagged compared to the inflow conditions. The cell perturbation method is definitely a promising approach, which we have now mentioned in the text. In fact, it would be worth implementing in PALM to enable a direct apple-to-apples comparison for evolving boundary conditions, which we mention in the revised summary and conclusions section.

*39) Page 43, line 20. This is likely caused by the lateral boundary conditions that do not match mesoscale variability (i.e., they are uniform in space and do not change between the LES domain boundaries).*

Indeed, horizontally averaging the boundary conditions does change the inflow characteristics. The attached Figs. 1 and 2 show the boundary layer structure with homogeneous and heterogeneous boundary conditions, respectively. Using heterogeneous boundary conditions we do see reduced differences during the morning transition, while the boundary layer remains similarly heterogeneous during the afternoon transition. So the homogenized boundary conditions can not fully explain the phenomenon. Since we do see

an improvement for this case in the morning transition, we relativized our suggestion to prefer homogeneous boundary conditions at the end of section 4.4.

**Response to Reviewer 2**

**Major comments:**

*1.) In a nested model setup, a sponge zone or a damping layer is commonly adopted at the lateral boundaries of the nest, such that spurious wave reflections due to changes in grid resolution across the nest interface are absorbed. Correct me if I am wrong, but in the proposed implementation, Dirichlet boundary conditions is adopted to drive the PALM model. Would this cause numerical issues at the lateral boundaries, especially the outflow boundary? It would be hard to observe spurious reflections at the lateral boundaries under convective conditions, it would be much easier to spot numerical waves under nighttime stable conditions. Also, could the absence of a sponge layer at least partially explain the "rim" of time-averaged vertical motions along the lateral boundaries presented in Fig. 19? If implementation of a sponge zone is out of the scope for the current model, the authors should at least provide some justification for using Dirichlet lateral boundary conditions.*

In the current implementation, we set Dirichlet boundary conditions at the five open model boundaries, which is the natural choice in our opinion. We had, in fact, looked into this during the implementation of the mesoscale nesting. Even though especially under nighttime stable conditions the boundary-induced vertical motions cause gravity waves that propagate into boundary-normal direction, we have not observed any reflections of gravity waves at the boundaries. They may still exist but are probably masked by the flow-adjustment effects near the boundaries so that reflections do not become apparent in the mean flow fields.

Other models such as WRF employ a damping zone of 5 grid points near the lateral inflow/outflow boundaries where the fine-grid solution is relaxed with respect to the coarse grid solution according to Davies and Turner (1977), mainly to damp horizontally propagating waves. During the implementation of an LES-LES nesting capability, which is described by Hellsten et al. (2020), we have experienced problems with relaxation of the child solution towards the parent solution and vice versa. This was attributed to a not fully conservative interpolation which finally led to strange acceleration/deceleration effects within the nested child. As we did not observe any issues with wave reflection at the lateral boundaries, neither in this mesoscale one-way nesting implementation nor in the one-way and two-way LES-LES nesting, we did not follow this approach.

Nevertheless, the reviewer raised a legitimate point, i.e. if the near-boundary up/downdrafts can be smoothed with a proper relaxation. For this reason we ran a test simulation where we considered a relaxation zone of 5 and 25 grid points, similar to the formulation in WRF. The attached Figs. 3 and 4 show an xy-cross-section of the mean vertical speed with and without a 25-gridpoint-wide relaxation zone (5-grid-point wide relaxation test looks pretty much the same). In this case, the up- and downdrafts at the inflow and outflow boundaries, respectively, are of similar strength as without any relaxation (please compare to Fig. 19 in the manuscript). Hence, according to this test, we think that a sponge layer will not help much with respect to this boundary-induced vertical motions, though it also does not do any harm with respect to the turbulence generation: The turbulence adjustment in TKE and skewness look pretty much the same. We have made a note in the text that we go without implementation of a damping zone.

The main difference between PALM and mesoscale models such as WRF is that PALM is a pure incompressible formulation at the moment. With this, acoustic waves are already filtered out and

we can only resolve gravity waves. We are not entirely sure about the initial motivation to implement such damping zones, but in case of a compressible system this might become important to get rid-off wave-reflection.

With regard to Fig. 19 and the attached one, please note that we had a bug in our original setup which caused the slight updraft towards the north-west in the original Fig. 19. We have fixed this and re-run all the simulations and re-plotted all figures. The changes were almost invisible in all plots, except for Fig. 19 where this slight updraft in the north-west has vanished.

*2.) Does COSMO have LES capability? If so, is the LES closure in the COSMO model the same as that in the PALM model? If the answer is also yes, I would suggest the authors try the following experiment, to help diagnose some of the issues such as the mismatch of potential temperature profiles and wind profiles. The authors could set up a similar LES domain within the COSMO model, and run the same test case within COSMO. (I assume COSMO has one-way nesting capability). The differences due to the model coordinate systems should not matter too much given the limited horizontal extent of the LES domain. Then the authors could compare the nested COSMO results with the nested PALM results to understand, for example, the influence of different land- surface schemes on the vertical wind and temperature profiles in the nested domain.*

Indeed, there is an LES-like subgrid-scale turbulence parameterization available in COSMO by Herzog et al. (2002a, 2002b) and we agree, such a comparison would be insightful. Unfortunately, we are not experienced with the COSMO model system and so far have only used the operational model output by DWD. Thus, we currently can not justify the resources to carry out such an analysis. In any case, as we discussed in our introductory remark, such an analysis would fall outside of the scope of the present manuscript.

**Minor comments:**

*1. Page 3, Lines 25-26, I have to disagree with this statement. Both the synthetic turbulence method and the cell perturbation method add "artificial" perturbations to the flow that are not strictly consistent with the "physics of turbulence production". Also, I believe the cell-perturbation method is also capable of allowing turbulence to "freely develop depending on the mean-gradients of potential temperature and wind speed". I would like to hear your explanation but I don't think one method has an advantage over the other in terms of physics.*

The reviewer is right. As we have outlined in our reply to reviewer 1's comment 3, both approaches do not strictly match the physics of turbulence generation, so that we have removed the discussion from the text as it is misleading. Adding temperature fluctuations, as done in the cell perturbation method, first produces turbulence in the vertical velocity component which then is transferred towards the horizontal components via the pressure-correlation term. This does not match the physics in shear-driven boundary layers. Xie and Castro's (2008) approach, on the other hand, deviates from the physics in buoyancy-driven boundary layers by creating perturbations in the vertical momentum leading to temperature fluctuations, while in nature it would be the other way around. Moreover, both approaches only add perturbations on fixed scales. In the cell perturbation method, the perturbation scales arise from numerical

considerations, while in Xie and Castro's method perturbations show dominant spatial scales which do not necessarily reflect reality. However, turbulence covers a wide range of scales, which both methods do not account for.

*2. Page 5, please combine the first paragraph with the last single-sentence paragraph on Page 4.*

Done.

*3. Page 5, line 5, "Both nesting features may, however, may be . . .", please fix the grammar.*

Done.

*4. Page 7, line 10, better "do not generally" than "do generally not".*

Done.

*5. Page 7, lines 9-17, so the divergence is removed at the LES domain level, rather than at each grid point, is that right? Perhaps point this out explicitly.*

Yes, in other words, this correction needs to be applied only to make the boundary conditions compatible with the incompressible continuity equation. Also, there is generally no need for a local correction in incompressible models, since they already contain one in the form of the pressure correction step. We added a remark on this below Eq. 2.

*6. Page 18, perhaps the authors will explain later, but how is Eq. 17 implemented near boundaries, where points outside the computational domain are required in the double summation?*

We think the reviewer refers to Eq. 27 in the original manuscript. Yes, especially for larger integral length scales, random numbers from outside of the domain are required. To fulfill this, the required random numbers are also computed for locations outside of the domain and the respective arrays are allocated sufficiently large. We have noted this in the revised manuscript within the more technical paragraph about the parallelized implementation (3rd. paragraph below Eq. 27).

*7. Page 19, lines 14-15, is there a reason why "at opposite boundaries (west and east, as well as north and south) we use the same $\Psi i$" ?*

Yes, by doing this we want to save computational resources. As the computation of the perturbations, e.g. at the west boundary, is mainly parallelized along x (please see also our response to your comment 8), we have the same set of random numbers and perturbations available also along the east boundary. Though the imposed turbulence at the west and east boundary is identical, we think that this does not cause any harm since the boundaries are sufficiently far apart from each other so that no interaction occurs.

*8. Page 19, lines 22-23, correct me if I am wrong, but 2d domain decomposition means that the domain is split in the x and y directions, right? So why would this enable Eq. 27 to be computed locally, so that "no global communication is necessary". For example, on the west boundary, the summation still requires information across processors in the y-direction, right?*

The reviewer is right, the phrasing "so that no global communication is necessary" is misleading and we have revised the text accordingly. PALM uses a symmetric 2-d decomposition along the x- and y-directions. For example, the western boundary represents a y-z-layer, where the y-direction is treated by ny MPI ranks. For each of the ny MPI ranks a subpart of perturbations is required, i.e. a subset of the y-z layer. The computation of this subset of perturbations is then distributed among all nx MPI ranks along the x-direction. To do this, the same set of random numbers is generated on each of the respective MPI ranks along the x-direction, while each mpi-rank along x computes another yz subset of filtered perturbations. Finally, computed perturbations are gathered on the western boundary from all respective nx MPI ranks, so that for each yz location a value is available. For this, only communication among MPI ranks along x is required. The only exchange along the y-direction is required for computing the variances and the averages used for normalization to zero mean and unit variance. Since this is not really a global communication but only a one-dimensional communication pattern in x- and y-direction, we agree with the reviewer that our original phrasing was not correct.

*9. Page20, lines 24-25, so these parameterized Reynolds stresses apply only to un- stable conditions? What about stable and/or neutral conditions? Did you drop the MO term?*

Yes, in stable/neutral situations, the first term within the brackets is omitted. We have mentioned this explicitly in the revised text.

*10. Page 21, correct me if I am wrong, but zi is obtained by horizontal averaging along the "boundary grid point", but u* is obtained by horizontal averaging "within the model domain", why the difference?*

In the optimal case, we would get all information from the mesoscale model. However, depending on the mesoscale model used, as well as depending on the specific dataset, this information may not be available. While the boundary-layer depth can be estimated by the mean wind and temperature profiles, this is not possible for quantities that depend on surface properties. In order to generate turbulence that roughly matches the conditions within the LES domain, we have decided to infer surface-related quantities from horizontal averaging the LES generated values. Please see also our response to reviewer 1's comment 24. We clarified this in the revised manuscript.

*11. Page 22, Eq. 37, the turbulent time scale appears to be a dimensionless number, how is it transformed into actual time?*

We made an error in the equation; thanks for pointing this out. The turbulent timescales implemented are not dimensionless. We have revised the equation accordingly.

*12. Page 24, Line 12, "indicate" rather than "indicates".*

Done.

*13. Page 24, Line 25, "subsequently" rather than "in the subsequent",*

Done.

*14. Page 26, Line 7, you meant "RES"?*

We mistakenly used RES in some places in the manuscript where we intended to refer to the REF simulation. In the revised version, we consistently use REF. In particular, we changed RES to REF in Fig. 9 and its description.

*15. Page 26, Line 18, how do you set the prescribed values of H0 and LE0 ?*

In the test simulation PSF, we have taken the hourly COSMO sensible and latent heat fluxes and prescribed them as the lower boundary condition for the temperature and mixing-ratio balance equation, instead of calculating the fluxes from the coupled land-surface model. Therefore, we have divided the heat fluxes as shown in Fig. 10 by the density and cp (lv), respectively, in order to obtain fluxes in kinematic units as required in PALM. We have revised the sentence accordingly.

*16. Page 32, Line 13, I would intuitively expect a monotonic increase of resolved TKE from the coarse grid to the fine grid, approaching some asymptotic values inside the LES domain. But why the TKE peak?*

For neutral boundary layers, we expect the same behavior as the reviewer said. In the convective case, this peak in TKE is a result of missing entrainment fluxes which have not been fully developed right behind the boundary. Behind the inflow boundary the imposed perturbations trigger the formation of upward-rising thermals that can accelerate more or less unhindered until they reach the inversion layer where they experience negative buoyancy and start to entrain warmer air from above. This entrainment of warmer air from the free atmosphere slightly stabilizes the upper part of the boundary layer and causes the formation of wide and weak downdrafts, which then counteract the acceleration of the thermals. The TKE peak is, thus, a result of insufficient representation of the important boundary-layer processes near the inflow boundary. In fact, this is a well known behaviour in LES for convective boundary layers, which can also be observed in idealized simulations with e.g. cyclic conditions, where the domain-averaged TKE also shows some overshooting during the model spin-up phase right at the moment when the bulk of the initially uprising thermals reach the inversion layer. We have added a note on this in the text.

*17. Page 40, Lines 5-8, this are most likely compensating vertical motions due to horizontally divergent and convergent flow at the inflow and outflow boundaries, as a result of continuity.*

*I would not over-interpret this like "horizontal momentum needs to be transported downward forcing the flow to descend".*

We agree and simplified the sentence in the revised manuscript.

*18. Page 41, Fig. 20, please double check the legends, both "distributed" and "abso- lute" are marked with solid black lines.*

We agree that the legend without any explanation is a bit confusing. We had revised the legend and listed every shown graph explicitly.

*19. Page 41, line 2, "is" rather than "means".*

Indeed, "means" sounds a bit weird here. We now use 'represents'.

**Response to Interactive Comment**

*The parent domain (COSMO) has a resolution of 2.8 km and 2.2 km, while the LES nested domain(PALM) is of the resolution 25 m. In the case of WRF, the ratio is opti- mised to be 3:1. Is there any such condition for PALM:COSMO ?*

The mesoscale nesting of a PALM simulation in a larger-scale model such as COSMO does not depend on the grid-aspect ratio. This is because our nesting approach is purely one way. Indeed, the self nesting in WRF is limited by several constraints, as is the PALM internal LES-LES nesting, where the grid-aspect ratio can only be a multiple integer. This constraint in WRF's and PALM's internal nesting arises from numerical considerations where the interpolation from the coarse to the fine grid and back from the fine to the coarse grid, needs to be conservative (for further details please see Hellsten et al., 2020). If this conservative interpolation is not realized spurious numerical issues may arise. This conservative interpolation, however, is only realized for specific grid aspect ratios in the self nesting. In the COSMO-PALM nesting, which is a pure one-way nesting, such constraints do not exist and we just need to take care that the boundary conditions satisfy the incompressible divergence constraint (see our Eq. 2 and our answer to reviewer 2's minor comment 5).

**References**

Blay-Carreras et al., 2014: Role of the residual layer and large-scale subsidence on the development and evolution of the convective boundary layer, Atmos. Chem. Phys., 14, 4515–4530

Ching et al. 2014: Convectively Induced Secondary Circulations in Fine-Grid Mesoscale Numerical Weather Prediction Models, Monthly Weather Review, 142, 3284–3302

Davies, H. C., and R. E. Turner, 1977: Updating prediction models by dynamical relaxation: An examination of the technique. Quart. J. Roy. Meteor. Soc., 103, 225–245.

Hellsten A., K. Ketelsen, M. Sühring, M. Auvinen, B. Maronga, C. Knigge, F. Barmpas, G. Tsegas, N. Moussiopoulos, and S. Raasch, 2021: A Nested Multi-Scale System Implemented in the Large-Eddy Simulation Model PALM model system 6.0, Geosci. Model Dev., accepted

Herzog, H.-J., U. Schubert, G. Vogel, A. Fiedler, and R. Kirchner (2002a): LLM - the high-resolving nonhydrostatic simulation model in the DWD-project LITFASS. Part I: Modelling technique and simulation method. COSMO Technical Report No. 4, Deutscher Wetterdienst.

Herzog, H.-J., G. Vogel, and U. Schubert (2002b): LLM - a nonhydrostatic model applied to high-resolving simulations of turbulent fluxes over heterogeneous terrain. Theor. Appl. Climatol., 73, 67–86.

Lenschow DH, Mann J, Kristensen L (1994): How long is long enough when measuring fluxes and other turbulence statistics. J Atmos Ocean Technol 11:661–673

Stull, 1988: Introduction to boundary layer meteorology.

Weil et al. 2004: The use of large-eddy simulations in Lagrangian particle dispersion models. J Atmos Sci 61:2877–2887

Zhou et al. 2014: The Convective Boundary Layer in the Terra Incognita, Journal of the Atmospheric Sciences, 71, 2545–2563

---

## Referee Report (RR1)

**Mesoscale nesting interface of the PALM model system 6.0 (by Kadasch et al.)**
**Manuscript ID: GMD-2020-285**

This manuscript describes a new nesting interface for the high-resolution LES model PALM, using model output from the mesoscale model COSMO ($\Delta x = 2.8\,\mathrm{km}$) as boundary data, which leads to realsitic synoptic forcing. The authors describe the nesting interface "INIFOR" in a detailed, but concise way. The performance of INIFOR is shown with a case study over grassland in northeastern Germany with the simulation of an evolving daytime boundary layer. The interface performs sufficiently well, however, the LES performance partly suffers from the poor representation of boundary-layer processes in the mesoscale model. The authors reflect on these issues in detail and already suggest possible solutions to this problem, therefore this manuscript can be published in Geoscientific Model Development after very minor revisions.

**Specific comments**

- Page 8, lines 3-4: Please state that turbulence in COSMO is fully parametrized *at the current grid spacing ($\Delta x = 2.8\,\mathrm{km}$)*.

- Page 24, line 4: The geographical coordinates provided lead to Berlin's city centre (when entering them in Google Maps), but it is stated that the domain lies east of the city. It would, however, make sense to add the exact coordinates in the manuscript to match the description of the domain (grassland land surface type).

- Page 24, lines 13-14: It should be clarified that the convective rolls stem from the convection grey-zone, and not from the grey zone of turbulence (Wyngaard 2004).

- Figure 11: Please adjust the colorbar of the contourlevels to the same range of $292\,\mathrm{K}$ to $296\,\mathrm{K}$. This makes the figures more comparable and illustrates the diurnal cycle of the ABL in a better way.

- Figures 12 and 18: The colorbars in these figures are somewhat misleading, because zero is not in the middle. Please readjust them.

**References**

Wyngaard, J. C., 2004: Toward Numerical Modeling in the "Terra Incognita". *J. Atmos. Sci.*, **61 (14)**, 1816–1826, doi:10.1175/1520-0469(2004)061⟨1816:TNMITT⟩ 2.0.CO;2.

---

## Author Response (AR2)

**Mesoscale nesting interface of the PALM model system 6.0 — Author response**

We thank both referees for their review of our revised manuscript. While both referees accept the manuscript for publication, referee 3 recommends a few minor technical changes before publication. We address those below, noting the referee's comments in *italics* and our responses in regular type.

Finally, we want to extend our thanks once more to all referees for their thorough analysis of our manuscript and their valuable suggestions to improve it.

Best regards,

Eckhard Kadasch, on behalf of all coauthors

**Response to Referee 3**

*1) Page 8, lines 3-4: Please state that turbulence in COSMO is fully parametrized at the current grid spacing ($\Delta x$ = 2.8 km).*

Since COSMO's horizontal grid spacing depends on the particular configuration, we added a mention of "horizontal grid spacings of several kilometers".

*2) Page 24, line 4: The geographical coordinates provided lead to Berlin's city centre (when entering them in Google Maps), but it is stated that the domain lies east of the city. It would, however, make sense to add the exact coordinates in the manuscript to match the description of the domain (grassland land surface type).*

Thanks for pointing this out. We have corrected the values in the text. In earlier versions of our setup, we have used these coordinates, but later on we decided to move the model domain towards the east where our assumed homogeneous grassland fits much better with the true vegetation. The origin of the PALM domain used in our benchmark simulation is located at 52.5°N and 13.7°E which, indeed, is located east of Berlin.

*3) Page 24, lines 13-14: It should be clarified that the convective rolls stem from the convection grey-zone, and not from the grey zone of turbulence (Wyngaard 2004).*

We clarified this in the text.

*4) Figure 11: Please adjust the colorbar of the contourlevels to the same range of 292 K to 296 K. This makes the figures more comparable and illustrates the diurnal cycle of the ABL in a better way.*

We agree with the referee that a unified colorbar would illustrate the diurnal cycle of the ABL better. However, our main goal with this particular figure was to highlight the horizontal heterogeneity of the ABL due to mesoscale forcing. With a unified colorbar for all three panels, this information would partly get lost (as we show in the attached panel plot with a unified colorbar for 10 UTC, 13 UTC and 16 UTC). For this reason, we would like to keep the plot with separate colorbars. To avoid misinterpretations, we now explicitly mention and explain the reason for the different temperature ranges in the figure caption in our revised manuscript.

*5) Figures 12 and 18: The colorbars in these figures are somewhat misleading, because zero is not in the middle. Please readjust them.*

While the zero mark was in the middle, the positive and negative axes in these figures did not share the same scale. That is, reds and blues of the same intensity represented different velocity magnitudes. In the revised figures, we adjusted the colorbars such that this is no longer the case and similar shadings in red and blue represent the same magnitude. Since updrafts in our plots are sharper and stronger than downdrafts, we extended the colorbar on the positive side to darker shadings.